# Historical and current spatiotemporal patterns of wild and vaccine-derived poliovirus spread

Darlan da Silva Candido ®[1] ✉, Simon Dellicour ®[2,3,4], Laura V. Cooper[1], Carlos A. Prete Jr[1,5], David Jorgensen ®[1], Christopher B. Uzzell[1], Arend Voorman[6,7], Hil Lyons ®[6,7], Dimitra Klapsa[8], Manasi Majumdar[8], Kafayat Arowolo ®[8], Corey M. Peak[6], Ananda S. Bandyopadhyay[6], Javier Martin[8], Nicholas C. Grassly ®[1] & Isobel M. Blake ®[1]

Outbreaks of vaccine-derived poliovirus type 2 (cVDPV2) have become a major threat to polio eradication. However, variations in spatiotemporal spread have not been quantified. Here we analysed cVDPV2 cases and wild poliovirus type 1 sequences to uncover spatiotemporal patterns and drivers of poliovirus spread. Between 1 May 2016 and 29 September 2023, 3,120 cVDPV2 poliomyelitis cases were reported across 75 outbreaks in 39 countries. Outbreaks had a median observed circulation of 202 (range 0–1,905) days and a median maximum distance of 231 (range 0–4,442) km. Wavefront velocity analysis of large outbreaks revealed a median velocity of spread of 2.3 (5th–95th percentile 0.7–9.2) km per day. International borders were associated with a slower velocity of spread (P < 0.001), in periods with high estimated population immunity. Phylogeographic analysis of 1,572 global wild poliovirus 1 sequences revealed that historic spread resembles recent cVDPV2 patterns and that international spread is largely sustained by unidirectional movement between neighbouring countries. Our findings offer insights for enhancing the geographical scope of vaccination response in the final phases of poliovirus eradication.

After decades of intense poliovirus vaccination efforts, eradication of wild types 2 and 3 was declared in 2015 and 2019, respectively, while the spread of type 1 wild poliovirus (WPV1) has now been contained to only two endemic countries (Pakistan and Afghanistan)[1,2]. However, one of the main threats to achieving the eradication of poliovirus is the emergence and spread of vaccine-derived poliovirus (VDPV), especially serotype 2 (VDPV2), which has been causing growing numbers of acute flaccid paralysis (AFP) cases globally[3]. As such, the World Health

Organization (WHO) classifies the spread of poliovirus as a public health emergency of international concern[4].

Historically, two vaccines were available to immunize against poliovirus, the widely used live-attenuated Sabin oral poliovirus vaccine (OPV) and the inactivated poliovirus vaccine, used mostly in high-income countries until 2016 after which at least one dose has been incorporated into routine immunization programmes worldwide. OPV is cheaper, easier to administer and generates a mucosal

[1]MRC Centre for Global Infectious Disease Analysis, School of Public Health, Imperial College London, London, UK. [2]Spatial Epidemiology Lab, Université Libre de Bruxelles, Brussels, Belgium. [3]Department of Microbiology, Immunology and Transplantation, Rega Institute, Laboratory for Clinical and Epidemiological Virology, KU Leuven, Leuven, Belgium. [4]Interuniversity Institute of Bioinformatics in Brussels, Université Libre de Bruxelles, Vrije Universiteit Brussel, Brussels, Belgium. [5]Department of Parasitology, Institute of Biomedical Sciences, University of São Paulo, São Paulo, Brazil. [6]Gates Foundation, Seattle, WA, USA. [7]Institute for Disease Modeling, Gates Foundation, Seattle, WA, USA. [8]Division of Vaccines, Medicines and Healthcare products Regulatory Agency, South Mimms, UK. ✉e-mail: ddasilva@ic.ac.uk

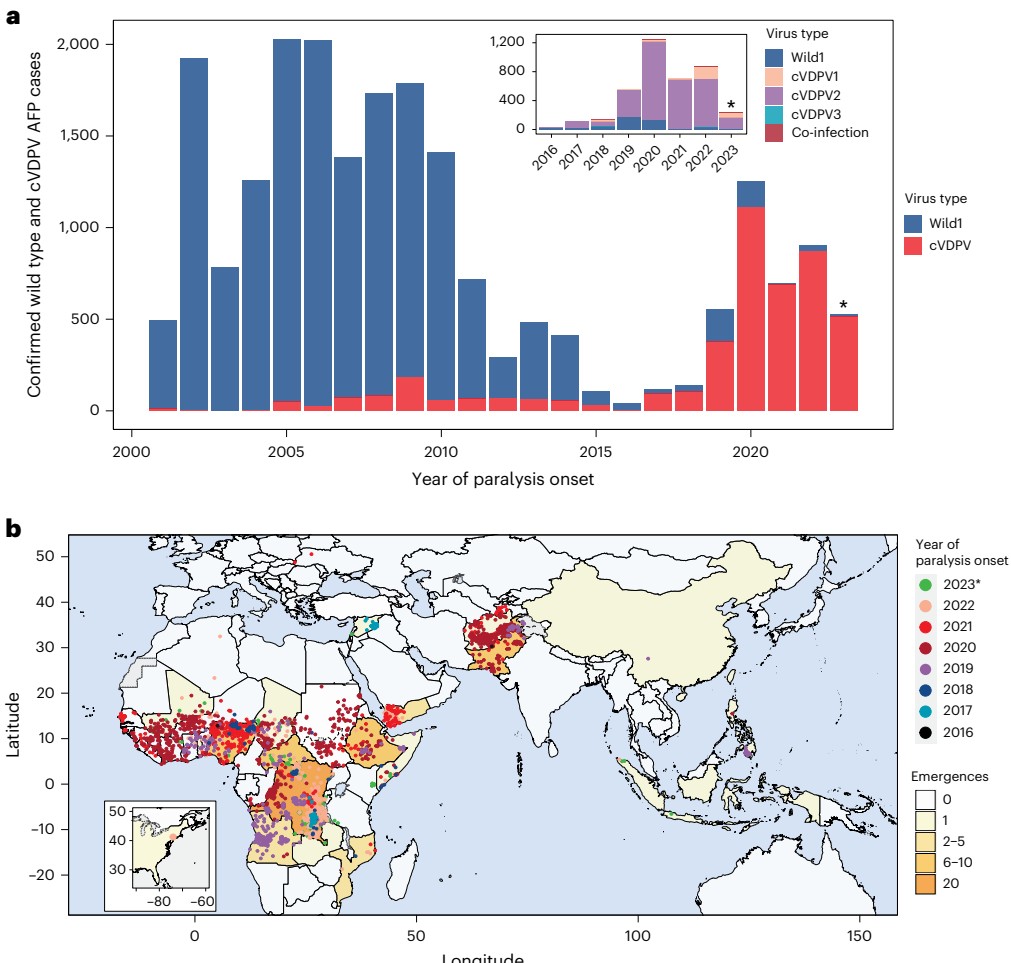

**Fig. 1 | The global spread of cVDPV2 outbreaks. a**, Timeseries of poliomyelitis cases from 2001 to September 2023 according to virus type: wild type 1 (dark blue) and VDPV (red). The inset depicts the distribution of cVDPV cases according to serotype: cVDPV1 (beige), cVDPV2 (purple), cVDPV3 (light blue), WPV1 (dark blue) and co-infections (dark red). Asterisks highlight years with incomplete data, that is, until September. **b**, Global distribution of cVDPV2 cases between May 2016 and September 2023. Points are coloured by date of onset and countries are coloured by number of VDPV2 emergences (outbreaks first detected in that country). Asterisks highlight years with incomplete data, that is, until September.

response against poliovirus infection and transmission[5]. However, it is also genetically unstable[6]. In rare instances, vaccine strains can acquire mutations associated with paralysis and transmissibility and subsequently lead to outbreaks caused by circulating VDPV (cVDPV)[3,7,8]. In 2016, the global poliovirus eradication initiative (GPEI) discontinued routine immunization of children with trivalent OPV (serotypes 1, 2 and 3) and replaced it by its bivalent form (serotypes 1 and 3). 'The Switch', as the replacement is called, was aimed at stopping the use of OPV2 and consequently the risk for new emergences of cVDPV2 outbreaks, and was justified by the certification of WPV2 eradication[9]. However, low vaccination coverage at withdrawal[10] and the use of monovalent Sabin OPV2 in response to cVDPV2 transmission persisting from before The Switch have generated multiple new cVDPV2 outbreaks[11,12]. Vaccination responses to these outbreaks were too small and slow to stop spread[10]. Since March 2021, a novel type 2 OPV (nOPV2), engineered to be more genetically stable, has also been largely used to respond to VDPV2 outbreaks under an Emergency Use Listing issued by the WHO, and has received subsequent licensure and prequalification since December 2023[13]. This has reduced the number of cVDPV2 emergences, but transmission of outbreaks persists in part owing to limited, delayed responses and a failure to prevent the geographic spread of the virus.

To achieve the goal of eradication, we need to better understand the spatiotemporal transmission of poliovirus to inform vaccination campaigns and policy. Here, we use data from 75 cVDPV2 outbreaks across 46 countries detected between The Switch, in May 2016, and September 2023 to quantify their variation, speed of spread and factors associated with these differences. In addition, we compare current cVDPV2 and historical WPV1 spatial spread through phylogeographic analysis of publicly available and previously unreported WPV1 genetic sequences. We discuss how these data can be used during vaccination campaign planning to maximise their impact and stop widespread transmission of cVDPV2 outbreaks.

## Results

### General characteristics of cVDPV2 outbreaks

Between 1 May 2016[14] and 29 September 2023, 3,898 poliomyelitis cases (that is, 436 WPV and 3462 VDPV cases) were reported across a total of 46 countries in all six WHO regions (African Region (AFRO), Region of the Americas (AMRO), Eastern Mediterranean Region (EMRO), European Region (EURO), South-East Asia Region (SEARO) and Western Pacific Region (WPRO); Fig. 1a). A total of 3,120 (80%) VDPV cases were linked to cVDPV2 circulation from 75 distinct outbreaks (Fig. 1b and Supplementary Table 1; see the Methods for outbreak classification). Since 2017, most of the reported poliomyelitis cases were caused by vaccine-derived strains (Fig. 1a). While WPV1 cases during the study period were only reported in five countries — mainly concentrated in Pakistan and Afghanistan, with some cases in

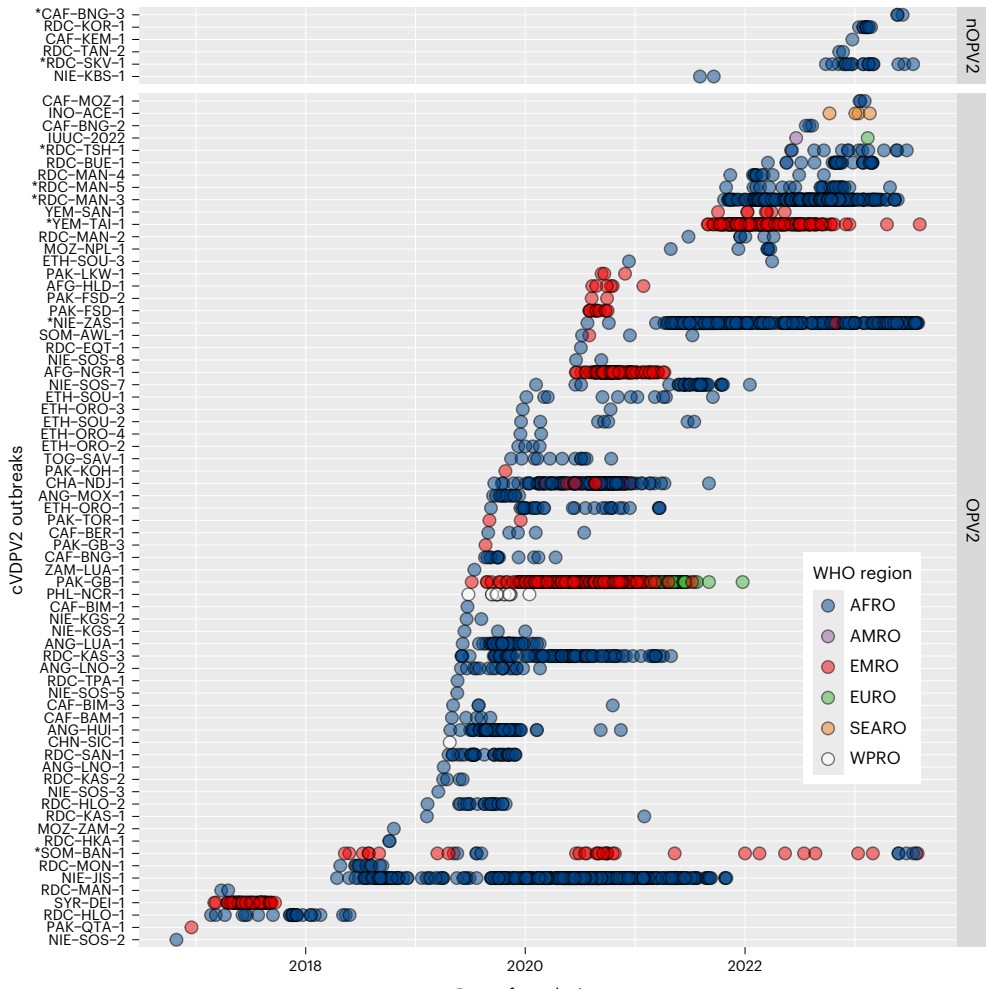

**Fig. 2 | Distribution of cVDPV2 cases globally per outbreak.** Temporal distribution of cases between May 2016 and September 2023 per cVDPV2 outbreak stratified according to the vaccine that seeded the outbreak: Sabin OPV2 or nOPV2. Asterisks highlight ongoing outbreaks (cases reported within the last 6 months, as of September 2023). Circles are coloured according to the WHO region of the cases: AFRO (dark blue), AMRO (purple), EMRO (dark red), EURO (green), SEARO (orange) and WPRO (white).

Malawi, Mozambique and Nigeria—cVDPV2 cases emerged and spread in most of Africa and areas of South Asia (Fig. 1b). In general, individual cVDPV2 outbreaks were relatively small, with a median size of 5 cases, but outbreak size distribution is skewed, ranging from 1 to 578 cases with a mean size of 41.6 cases (Fig. 2 and Supplementary Figs. 1 and 2). Importantly, given that the largest outbreak is still ongoing (cases detected in the last 6 months), this range will continue to increase. Of the 75 outbreaks, 19 (25%) have an estimated size ≥20 cases and only 6 outbreaks reported over 200 cases: NIE-ZAS-1 (578, ongoing), NIE-JIS-1 (428), PAK-GB-1 (393), RDC-MAN-3 (340, ongoing at the time of analysis), YEM-TAI-1 (221, ongoing) and CHA-NDJ-1 (220). As of September 2023, most cVDPV2 outbreaks emerged from Sabin OPV2 (70, 92.1%), while only six outbreaks (7.9%) were linked to the recently introduced nOPV2 (Fig. 2 and Supplementary Fig. 2). While the reasons for the variation in outbreak size are not entirely clear, delayed vaccination response after first detection is associated with larger outbreaks, with an increase of 12% per additional week to response[15].

## Spatiotemporal dynamics of cVDPV2 spread

Outbreaks emerged in 19 different countries and further spread to at least 22 other countries (Fig. 1b and Supplementary Table 1). Most outbreaks were first detected and probably emerged in the Democratic Republic of Congo (DRC, 20), followed by the Central African Republic

(9), Nigeria (9), Pakistan (9) and Ethiopia (8; Supplementary Fig. 3). The DRC (663), Nigeria (547), Afghanistan (351), Yemen (228) and Chad (182) have reported the most cases (Fig. 2b and Supplementary Fig. 4). In terms of international spread, most outbreaks (53, 71.6%) only spread locally in the country of first detection, and only eight (10.8%) outbreaks spread to at least two other countries: JIS-1 (14 countries), NIE-ZAS-1 (12), CHA-NDJ-1 (5), PAK-GB-1 (3), NIE-SOS-7 (2), RDC-SKV-1 (2), SOM-BAN-1 (2) and TOG-SAV-1 (2). It has recently been estimated that most cVDPV2 outbreaks between 2016 and 2019 would have probably been seeded by a small number of campaigns in Nigeria, Ethiopia and the DRC, with a large campaign size, low immunity and no follow-up campaigns as risk factors for new emergences[16,17].

Outbreaks that emerged in Nigeria have often, directly or indirectly, spread to neighbouring Niger (3), Chad (2), Benin (2) and their neighbours, Togo (2) and Mali (2). Two examples of this pattern of spread are NIE-JIS-1 and NIE-ZAS-1, which emerged in Jigawa and Zamfara states, respectively (Fig. 3a,b). However, while NIE-JIS-1 spread mostly outside Nigeria, NIE-ZAS-1 spread mostly within Nigeria with more limited transmission in other countries, in our period of analysis. Other examples of transmission between neighbouring countries are the Chad outbreak CHA-NDJ-1 with cases subsequently detected in the Central African Republic, Sudan, South Sudan and Eritrea and outbreaks in Angola, which have subsequently been detected in the DRC (Fig. 3c,d).

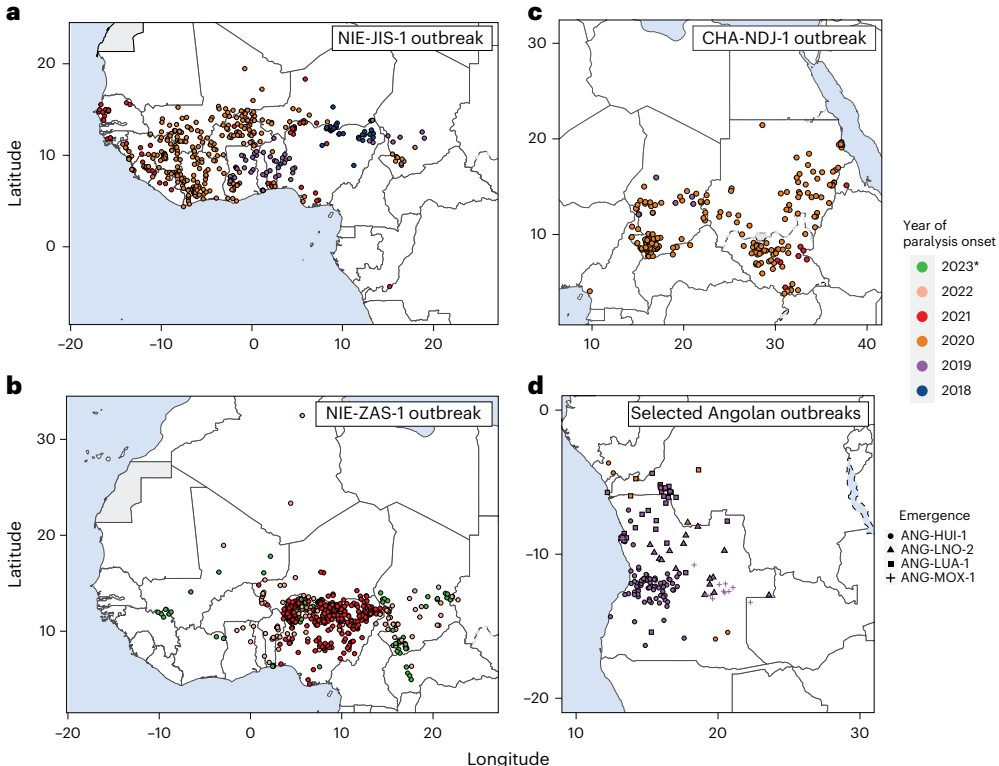

**Fig. 3 | Spatiotemporal distribution of reported poliomyelitis cases for selected cVDPV2 outbreaks. a,c,** Points represent each confirmed case and are coloured according to the year of symptom onset. The maps show the continuous flow of cases between neighbouring countries across different outbreaks and highlight the different spatiotemporal patterns between the NIE-JIS-1 (**a**) and the NIE-ZAS-1 (**c**) outbreaks, which both emerged in northern Nigeria and spread to neighbouring countries. **b,** Spatiotemporal distribution of cases for the CHA-NDJ-1 outbreak. **d,** Spatiotemporal distribution of cases for four selected Angolan outbreaks.

We also investigated the maximum observed distance of spread from the first reported case (origin) and the duration of cVDPV2 outbreaks as a proxy for timeliness and effectiveness of response. The spatial spread of cVDPV2 outbreaks has ranged between 0 km and 4,442 km (median maximum distance from origin: 231 km). The maximum distance each outbreak has spread varied according to country of first detection (Supplementary Fig. 5a) and the number of emergences, that is, countries with multiple emergences of cVDPV2 outbreaks presented lower median maximum spread distances than countries with single emergences (Mann–Whitney $U$ test, $P = 0.018$; Supplementary Fig. 6a). The median observed duration of cVDPV2 outbreaks was 202 (range 0–1905) days, with considerable variation between countries (Supplementary Fig. 5b) but no significant impact of multiple emergences per country (Supplementary Fig. 6b). When separating outbreaks by year of first AFP detection, both median duration and spatial spread reached their peaks during the coronavirus disease 2019 pandemic (2020 and 2021), with a tendency to decrease thereafter—but this should be interpreted with caution given the high proportion of ongoing outbreaks (Supplementary Fig. 7).

To investigate the relationship between time and distance in cVDPV2 outbreaks, we regressed the distance for the farthest case from their respective outbreak origin (first detected case) against the time of detection for large outbreaks—herein defined as 20 or more poliomyelitis cases. We found a strong positive correlation between distance and time (Pearson's $r = 0.79$, $P < 0.001$; Fig. 4a), suggesting that, overall, the longer an outbreak lasts the farther it spreads. A similar correlation was found when the same correlation analysis was performed for all cases of large outbreaks (Pearson's $r = 0.60$; Supplementary Fig. 8). However, the correlation between time of detection and distance varied considerably according to outbreak (Supplementary Fig. 9). Such variation can be partially explained by differences in the local population immunity levels at the time of spread. Outbreaks with a strong correlation between distance and time (Pearson's $r > 0.6$) were found to have higher estimated type 2 population immunity (median administrative region 2 (admin-2) immunity at the time of case onset of 62.8%)[18] (Supplementary Fig. 10a,b) when compared with outbreaks with lower or no correlation between distance and time (Pearson's $r < 0.6$, median immunity of 33.8%, Mann–Whitney $U$ test $P < 0.001$). This suggests a role for immunity in disrupting the correlation between outbreak duration and distance.

## Wavefront velocity of cVDPV2 outbreaks and the impact of international borders

We used a methodology previously adapted to study the wavefront dynamic of a local African swine fever outbreak to estimate the wavefront velocity of cVDPV2 outbreaks[19–21] and investigate factors that might impact it. A measure such as outbreak wavefront velocity can be used to better inform the geographical scope of vaccination campaigns on the basis of the time of onset and origin of reported cases. To this end, we interpolate the invasion times (date of paralysis onset) and the geographical distances of the cVDPV2 cases that actually extend the wavefront of each outbreak, as determined by successive daily kernel density polygons (Methods). The interpolated invasion times were then used to estimate a friction map (time/distance), and finally the outbreak wavefront velocity (distance/time; Supplementary Fig. 12). Our results show that cVDPV2 outbreak wavefronts move at a median speed of 2.3 km per day and a 95% quantile of 9.2 km per day across most outbreaks, with specific outbreak median wavefront velocities ranging between 0.9 km per day and 3.8 km per day. However, the distribution of

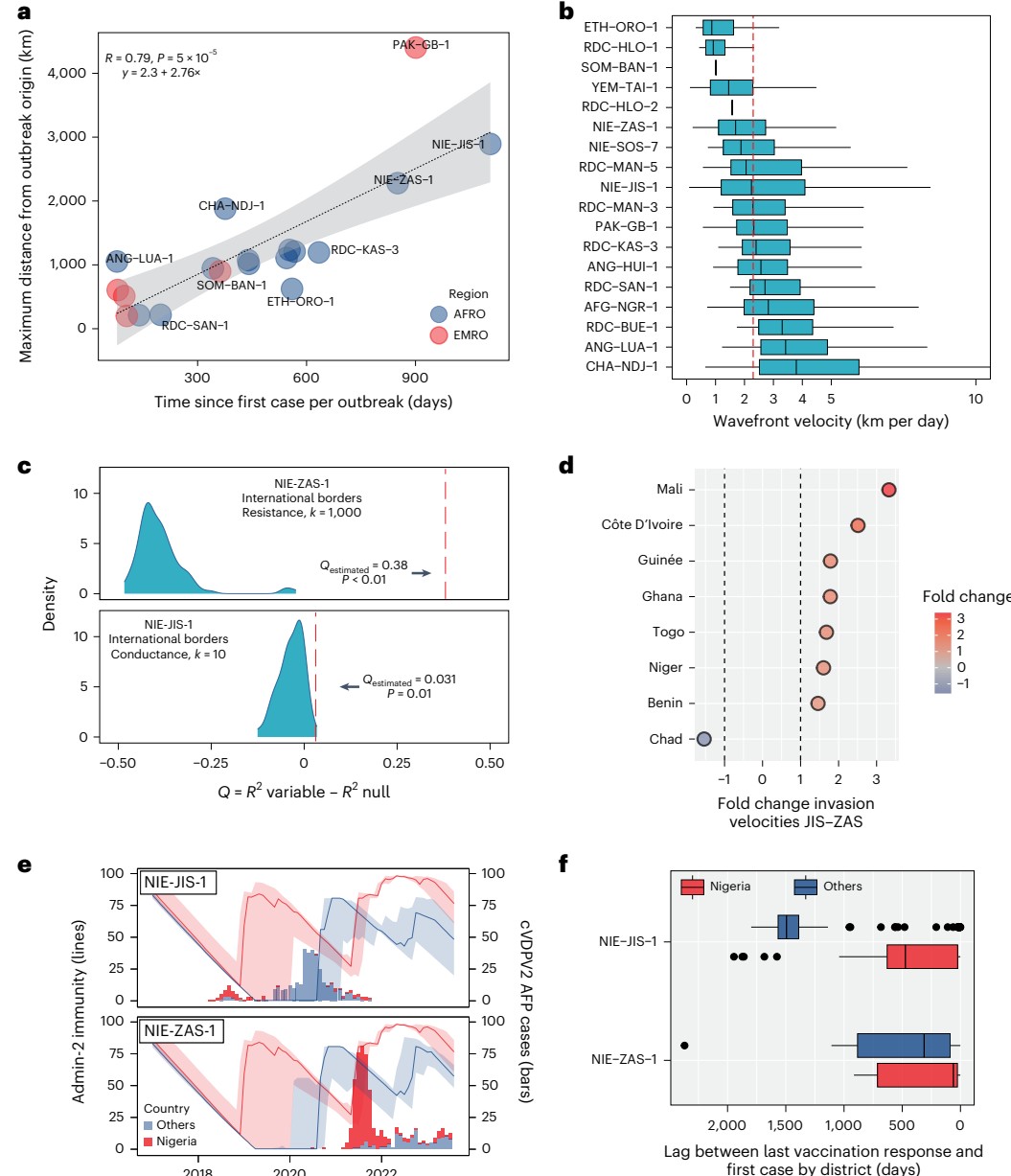

**Fig. 4 | Spatiotemporal characteristics and drivers of cVDPV2 outbreaks.**
**a**, Correlation between time (difference in days between the date of paralysis
onset for the case with largest distance from the outbreak origin and that of the
first case for each outbreak) and distance (distance from outbreak origin) for
all cVDPV2 cases of large outbreaks (≥20 cases). Dotted line and shaded area
represent the estimated linear regression line and the 95% confidence interval
(two-sided test). Correlation was assessed using a two-sided Pearson correlation
test. **b**, Box plot showing the distribution of wavefront velocity estimates for
large cVDPV2 outbreaks: NIE-ZAS-1 ($n$ = 578), NIE-JIS-1 (428), PAK-GB-1 (393),
RDC-MAN-3 (340), YEM-TAI-1 (221), CHA-NDJ-1 (220), AFG-NGR-1 (135),
RDC-KAS-3 (105), ANG-HUI-1 (79), ANG-LUA-1 (49), SOM-BAN-1 (40), NIE-SOS-7 (35),
RDC-SAN-1 (32), ETH-ORO-1 (28), RDC-HLO-1 (27), RDC-MAN-5 (26), RDC-BUE-1
(24) and RDC-HLO-2 (20). The central horizontal line indicates the median
(50th percentile). The bounds of each box represent the interquartile range
(IQR; 25th–75th percentiles). Whiskers extend to the most extreme data points
within 1.5 × IQR from the lower and upper quartiles. Whisker ends correspond
to the minimum and maximum non-outlier values. Outliers were removed
for ease of visualization. The full version is shown in Supplementary Fig. 11.
**c**, Impact of international borders on the wavefront velocity of the two largest
cVDPV2 outbreaks so far: NIE-ZAS-1 (578, ongoing, $P$ < 0.01) and NIE-JIS-1 (428,
$P$ = 0.01). Red dotted lines represent the actual estimated $Q$ for each outbreak.
$Q$ represents the proportion of the heterogeneity in the wavefront velocity that

can be associated with the tested variable (see Supplementary Information for
full details). Density plots depict the distribution of $Q$ values obtained under a
null dispersal model in which the tested environmental factors do not impact the
outbreak wavefront velocity. As detailed in Supplementary Information, such a
null dispersal model was obtained by a stochastic rotation of dispersal vectors
and tested using a one-sided permutation test. **d**, Fold change in the invasion
velocity of NIE-ZAS-1 over NIE-JIS-1. Invasion velocity for each outbreak was
estimated using the geodesic distance and the time of onset for the first case in
each country from each outbreak origin. **e**, The median (line) and IQR (shaded
areas) for the estimated vaccine-induced type 2 population immunity at admin-2
and timeseries of reported cVDPV2 AFP cases for Nigeria and all other countries
involved in the NIE-JIS-1 and NIE-ZAS-1 outbreaks, respectively. Lines and bars
are coloured according to location: Nigeria (red) and other countries reporting
cases from the same outbreak (blue). For details on immunity estimates, refer
to Supplementary Information. **f**, Time lag (days) between the first case
(date of onset) and the last vaccination campaign (campaign start date) per
district for each outbreak according to country group: NIE-JIS-1 Nigeria ($n$ = 38),
NIE-ZAS-1 Other (104), NIE-ZAS-1 Nigeria (192) and NIE-JIS-1 Other (221). The
central horizontal line indicates the median (50th percentile). The bounds of
each box represent the IQR (25th–75th percentiles). Whiskers extend to the
most extreme data points within 1.5 × IQR from the lower and upper quartiles.
Whisker ends correspond to the minimum and maximum non-outlier values.

wavefront movements varies considerably within outbreaks and tends to present a long tail, highlighting the importance of long-distance and fast-spread movements (Fig. 4b).

We then used the same framework to investigate the impact of population density, travel inaccessibility (minutes required to travel 1 m)[22] and international borders on the wavefront dispersal velocity. We did so by estimating the $Q$ statistic, which measures to what extent these variables would separately better explain the heterogeneity in wavefront dispersal velocities when compared with a null dispersal model[23] (Methods). We compared the NIE-JIS-1 and NIE-ZAS-1 outbreaks, as these are the largest cVDPV2 outbreaks so far; both started in northern Nigeria and spread to similar countries but have markedly different spatiotemporal patterns, as mentioned above (Fig. 3). For the NIE-ZAS-1 outbreak, we found that international borders and travel inaccessibility were negatively associated with the wavefront velocity by 38% ($P < 0.01$) and 6% ($P = 0.02$), respectively (Fig. 4c and Supplementary Fig. 13a), and we found no significant impact of population density (Supplementary Table 2). In particular, the wavefront progression of the NIE-ZAS-1 outbreak was thus notably associated with slowed international spread. Interestingly, for the NIE-JIS-1 outbreak, we found opposite results, that is, a small effect (3% each) of international borders and population density, both associated with increased spread ($P = 0.01$ and $P = 0.02$, respectively; Fig. 4c and Supplementary Fig. 13b), as well as no significant impact of the travel inaccessibility (Supplementary Table 2). This is supported by a median wavefront velocity that is 32% faster for the NIE-JIS-1 outbreak (Fig. 4b) and by a faster invasion velocity for NIE-JIS-1 in 87.5% (7/8) of the countries that reported both outbreaks (Fig. 4d).

A possible explanation for these contrasting results is that most of the NIE-JIS-1 outbreak happened in 2020, when the median estimated immunity in children aged 6–36 months[18] was high in Nigeria (median admin-2 immunity at symptom onset: 72.6%) but very low in all other countries involved in the outbreak (0.3%). Conversely, for the NIE-ZAS-1 outbreak, immunity was initially low in Nigeria and high in all other countries (in response to the ongoing NIE-JIS-1 outbreak). While immunity in Nigeria increased to high levels (84.8%), immunity in all other countries (65.5%) never decreased to levels as low as those during the NIE-JIS-1 outbreak (Fig. 4e). In fact, while the median time between the last vaccination campaign and the first NIE-JIS-1 case for Nigerian districts was of just over 1 year (473 days), for districts in other countries, the median lag was of over 4 years (1,494 days; Fig. 4f). For the NIE-ZAS-1 outbreak, the difference between the median lags for Nigerian districts and districts in other countries was much lower, 61 and 312 days, respectively. In summary, these observations suggest that the resistance effect of international borders on the wavefront dispersal velocity could potentially be attributed to population immunity, although not formally tested here (see limitations in the 'Discussion'), and that the impacts of population density and travel inaccessibility seem to be considerably more modest and context dependent. For all effects tested, see Supplementary Table 2.

### Historical and current routes for poliovirus spread

Given that limited cVDPV2 genetic data are publicly available since The Switch (Supplementary Fig. 14), we used a dataset of 1,572 WPV1 viral protein 1 (VP1) capsid-region genetic sequences, including 38 newly generated sequences from archived samples collected between 1953 and 2015 (Supplementary Table 3) to uncover historical routes of spread and compare them with the currently observed cVDPV2 patterns from case data. Historical phylogeographic reconstruction, using discrete trait analysis, identified at least four main clades on the basis of phylogenetic diversity, some with marked spatial subdivision (Fig. 5a). Considering that recent cVDPV2 outbreaks are concentrated in Africa and South Asia, we focused on clade 2 and 4 to understand the local patterns and routes of spread in these areas. At the country level, this analysis suggests Nigeria as the likely initial source of spread

(21 exports, 95% high posterior density interval (HPD) = 18–25) in the source–sink dynamics within clade 2 ($n = 201$, dating from 1985 to 2014; Fig. 5 and Supplementary Fig. 15). However, most exportation events from Nigeria happened to neighbouring countries (73.9%, 95% HPD = 73.0–76.0%), with some long-distance movements (directly or indirectly) into Ghana (26.1%, 95% HPD = 22.2–28.0%). In fact, this pattern of spread is consistent among countries in clade 2, importing and exporting from immediate neighbours (76%, 95% HPD = 74.3–77.6%). This suggests that, although Nigeria is the main source in the source–sink dynamics in clade 2, other countries are also important in establishing subsequent spread, such as Chad (6.5 exports, 95% HPD = 4–10), Ghana (3.3, 95% HPD = 3–5) and Angola (2.7, 95% HPD = 1–4). In terms of directionality, movement appears to be mostly unidirectional, with no pairs of exporter–importer countries being reversely identified in exporting–importing events. Exportation events from Nigeria to Cameroon and Chad, from Egypt to Sudan and from Angola to the DRC are also highlighted in phylogeographic analyses of clade 4b ($n = 559$, 1999–2011) and of 139 WPV3 VP1 sequences (1968–2012), reinforcing the transmission links between these countries (Supplementary Figs. 16–18). Such patterns, although being from different types of poliovirus and circulating over a different time period, are similar to patterns seen in the cVDPV2 AFP case data, with most movement happening between neighbouring countries.

## Discussion

The extent of current cVDPV2 transmission poses a huge challenge to complete polio eradication. The Switch has been a more difficult strategy than initially anticipated, and the epidemiology of poliovirus has changed after its adoption. Timely and sufficient scope of outbreak vaccination response is a key determinant of success in stopping outbreaks. Our findings carry important implications for response strategies and policy formulation. By estimating the spread velocity and common routes of spread, we can anticipate the geographical extent of an outbreak when planning vaccination campaigns, thereby expanding response coverage and curbing further dissemination. Currently, the standard operating procedures[24] from the GPEI relies on a reactive approach, with vaccination campaigns that follow the detection of cases in new populations (two rounds within 56 days of outbreak confirmation targeting at least 2–4 million children). However, exploring the potential cost benefits of proactive preventive campaigns in neighbouring countries with low immunity levels warrants further investigation. It is reasonable to speculate that the immunity generated in response to the NIE-JIS-1 outbreak may have initially mitigated the international spread of the NIE-ZAS-1 outbreak, akin to the effect of preventive vaccination campaigns and that, as such, international borders do not seem to prevent or even slow international spread. Thus, future vaccination campaigns aiming to interrupt poliovirus transmission should proactively account for spread between neighbouring regions, even in the absence of reported cases, and may use wavefront velocity estimates, including measures of variation (for example, the 90th percentile), to guide the geographical scope of interventions. These considerations remain pertinent, as the programme has transitioned to using nOPV2 (WHO prequalification status achieved in December 2023)[25], which appears to carry a lower risk of initiating new outbreaks on the basis of field data[26]. In addition, after our period of analysis, ongoing outbreaks have increased in size, especially NIE-ZAS-1, and new outbreaks have been detected[27].

Our analyses have some limitations. We have focused on analysing reported poliomyelitis cases through the AFP surveillance system. Many countries perform supplementary surveillance activities such as environmental surveillance and AFP contact or community sampling; however, these vary considerably across time and space and therefore we excluded these detections from our primary analyses. Second, given the limited number of cVDPV2 sequences publicly available since The Switch, our phylogeographic analysis focused

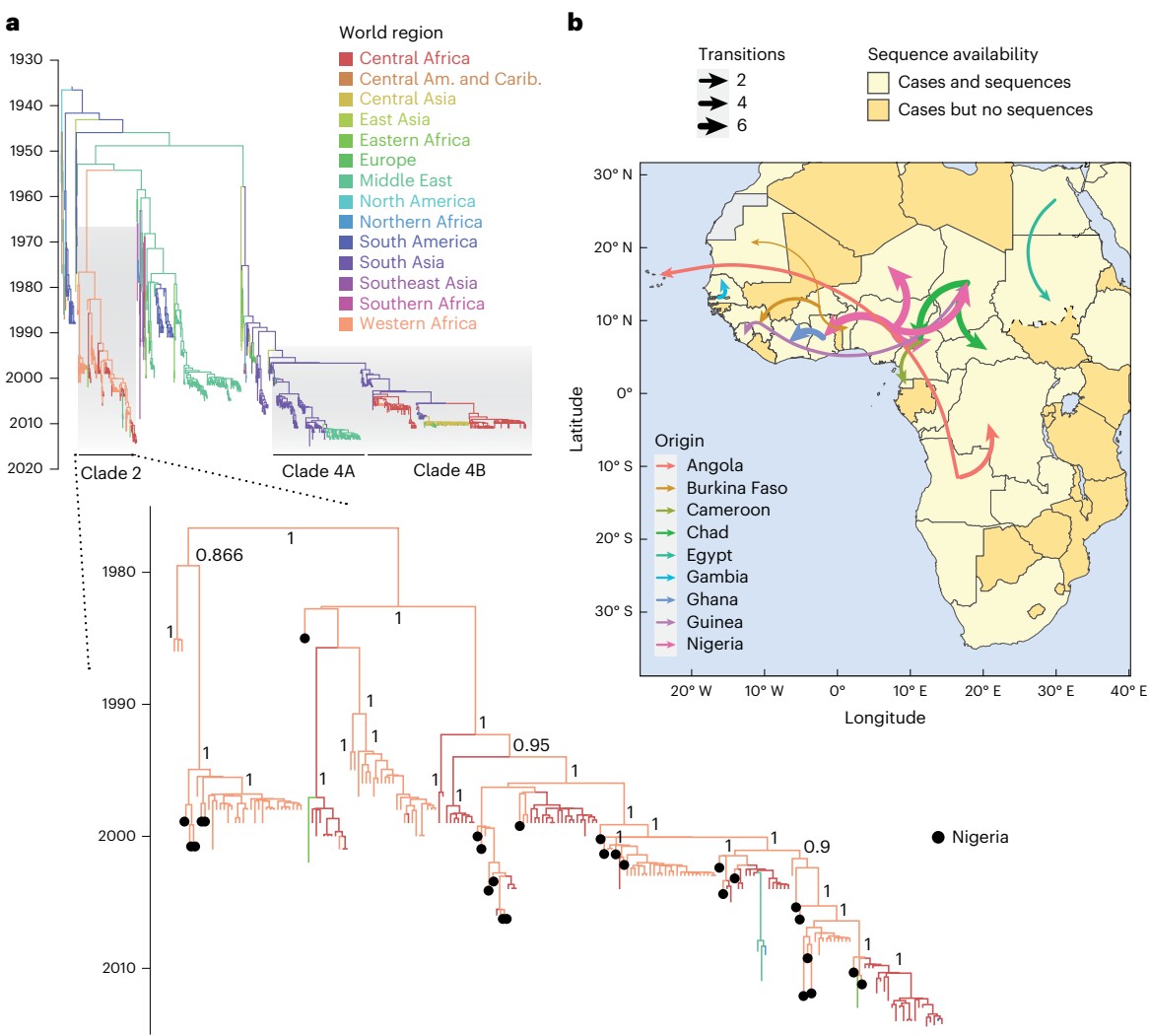

**Fig. 5 | Discrete phylogeographic analysis of 1,572 historical WPV1 VP1 sequences. a**, Time-rooted phylogeny of 1,572 sequences from cases and environmental samples downloaded from GenBank, and including 38 newly generated sequences, covering the period between 1958 and 2015. The country of collection was discretized according to 13 global regions, and clades were identified on the basis of the posterior node support and location support for internal nodes (Methods). Shaded areas highlight three clades of interest on the basis of the current patterns of cVDPV2 spread: clades 2, 4A and 4B. Expansion shows a sub-phylogeny for clade 2, which mainly represents the movement between Central (red) and West Africa (orange). Numbers on the main internal nodes of the clade 2 subtree (for readability) represent the Bayesian posterior support for each node. Black tips highlight sequences coming from Nigeria at the base of most clusters in that clade. A full version of this tree with all the posterior supports is shown in Supplementary Fig. 19. **b**, Sequence availability map and well-supported movements (Markov jumps, adjusted Bayes factor >20). Countries are coloured according to reporting of cases and availability of sequences: countries that reported cases and sequences (light yellow) and countries with cases but no sequences available (dark yellow).

on historical WPV1 sequences to shed light on similarities to current cVDPV2 spread. The Polio Information System (POLIS) contains information on outbreaks and emergence groups as well as mutation counts for each case but provides no access to the sequences themselves. As all VDPV cases were sequenced since the early 2000s, analysis of such data would provide a more accurate insight into the current patterns of cVDPV2 spread. It is important to acknowledge that the WPV1 sequences available for analysis represent ~0.2% of all reported WPV1 cases between 1980 and 2015 globally (0.06% between 1980 and 1999 and 5% between 2000 and 2015), and sampling is variable in space and time. We minimise spatial sampling heterogeneity by applying a tips-swapping approach (Methods; Supplementary Fig. 14). However, our models cannot account for unsampled areas, and we highlight that observed transmission links can be direct or indirect. In addition, given the limited availability of subnational information for the WPV1 sequences, we were not able to conduct a finer scale analysis of spread than at the country level. In addition, as

surveillance systems are often affected by underreporting, the true wavefront velocity might be faster than that estimated here. However, the true median wavefront velocity is probably within the bounds of the 90% quantile interval reported here, which should be considered when projecting poliovirus spread. Finally, owing to limitations in the software used for the wavefront analysis (Methods), which does not support time-varying covariates, population immunity could not be formally tested as a covariate using the wavefront velocity approach in our study. For this reason, spatiotemporal dynamics of immunity estimates have only been used as a potential explanation for the effect of international borders, and its potential role in poliovirus spread therefore warrants further investigation and formal testing.

While previous analyses by the GPEI have identified routes for WPV1 spread[28–30], our study highlights the additional insights provided by phylogeographic inference. Specifically, we demonstrate the capability of phylogeographic methods to estimate the number of movements between locations while considering sampling uncertainty.

Furthermore, while this manuscript does not delve into it owing to data constraints, phylogeographic techniques offer the potential to evaluate the effectiveness of intervention strategies, identify factors influencing virus lineage spread, compare the dissemination dynamics of different lineages and estimate measures of virus spread, as observed in studies on various other viruses[23,31–35]. These approaches hold substantial promise for enhancing the current GPEI strategic plan, particularly in increasing insight from surveillance data and improving response efforts. This is crucial not only for stopping the spread of cVDPV2 but also for addressing the remaining two serotypes of VDPV once all OPVs are withdrawn globally in the final phases of polio eradication.

## Methods

### Epidemiological data and outbreak definition

Institutional ethics approval for this study was granted by the Imperial College Research Governance and Integrity Team (reference no. 21IC6996). Epidemiological data on all AFP cases with paralysis onset after The Switch, 1 May 2016 through 29 September 2023, were downloaded from POLIS[36,37]. In brief, polio surveillance involves the surveillance of AFP cases, environmental samples, contact sampling and community sampling[38]. Poliomyelitis cases are laboratory confirmed according to WHO guidelines using virus isolation in cell culture, intertypic differentiation and genetic sequencing of the VP1 capsid protein[39]. Reported epidemiological data included information on, but were not limited to: dates of onset, notification, sample collection and reporting; administrative regions 0–2; geographic coordinates for each case—randomly sampled within the linked administrative region (admin-2) for data-protection reasons; and vaccination history.

A VDPV2 is determined as a virus containing at least six mutations in its VP1 region when compared with Sabin 2. An emergence group is defined as all phylogenetically linked isolates of a VDPV outbreak, that is, part of the same evolutionary and transmission process. These can include samples from AFP cases, the environment and healthy children (contacts and/or community). A VDPV2 outbreak, or emergence group, is determined by at least (1) two detections of genetically linked individuals (not necessarily AFP cases) that are not direct contacts, or (2) one AFP case and one environmental sample (ES; wastewater grab samples) or (3) two genetically linked ESs from different catchment areas or >2 months apart, if from the same area. VDPV2 isolates linked to outbreaks are called circulating VDPV2 (cVDPV2). As ES efforts vary considerably between countries, we decided to only investigate AFP cases in this study; thus, outbreaks with no positive AFP cases (ES-only) have not been included.

For the emergence group analysis (referred to as outbreaks in the main text), only confirmed cVDPV2 AFP cases with an associated emergence group were maintained in the dataset ($n = 3{,}120$). Although cVDPV2 sequences themselves are not available through POLIS, each case contains genetic information on the virus associated with the infection, such as number of nucleotide changes from Sabin 2 and the emergence group to which they are linked. Given that the data on POLIS do not contain detailed information on cVDPV2 emergences before 2010, for the 2001–2023 timeseries (Fig. 1), extranet WHO data on yearly case counts by country and virus type were downloaded from ref. 37.

Administrative boundaries were obtained from the WHO geodatabase and data on vaccination campaigns were obtained from POLIS. The publication of the maps in this manuscript does not imply the expression of any opinion whatsoever on the part of WHO concerning the legal status of any territory, city or area or of its authorities, or concerning the delimitation of its frontiers or boundaries.

### Spatiotemporal analysis of cVDVP2 poliomyelitis cases

Time of spread for each case was estimated by subtracting the date of paralysis onset by that of the observed first case in each outbreak.

Similarly, distance of spread was estimated by measuring the geodesic distance between each observed case and that of the first observed case in each outbreak, conducted in R[40] using the 'distGeo' function implemented in the 'geosphere' R package (version 1.5–18)[41]. In turn, the observed maximum outbreak duration was determined by the time of spread of the most recent observed case in each outbreak, while the observed maximum distance of spread was that of the most distant case from the outbreak origin (first observed case). It is important to highlight that, for outbreaks that were confirmed after the detection of both a single AFP case and a positive environmental sample genetically linked to that case, both the maximum distance of spread and the duration would be 0 if no further cases were reported (that is, 15 outbreaks in our dataset). To assess the correlation between time and distance of spread, we used outbreaks with a minimum of 20 observed cases as a measure of outbreak success and to reduce the effect of noise associated with the detection of cases in smaller outbreaks. The first case in each outbreak was also removed from the analysis to avoid inflating the correlation (that is, time and distance would both be 0 for all of them).

### Wavefront velocity analyses

In the absence of publicly available cVDPV2 sequences with information at the subnational level to perform a continuous trait phylogeographic analysis and infer the lineage dispersal velocity, we adapted a methodology previously used to estimate the wavefront velocity of African swine fever outbreak that occurred in 2018–2019 in the wild boar population in southern Belgium[19]. To estimate the wavefront velocity of cVDPV2 outbreaks, we used information on date of onset and the geographic coordinates of the centroid of the admin-2 polygon of origin for each cVDPV2 AFP case. We started our analysis by filtering only the cases that extended the outbreak wavefront. We did so by estimating 95% kernel density polygons for every day, $d$. For a given day $d$, we estimated the 95% kernel density polygon by considering the positions of all cases reported before $d$, and then used this series of daily polygons to discard infection cases that did not contribute to extending the outbreak area[19]. We used a 95% kernel density polygon mask to avoid extrapolation caused by outliers.

We then estimated both the observed dispersal distance and duration of each filtered case as the geodesic distance and the time in days, separating each filtered case from the first detected case of each outbreak, herein used as the most likely outbreak origin. To estimate and visualize the wavefront velocity on a map surface, we applied a previously described interpolation procedure[19,42,43] that uses a thin plate spline regression to interpolate the onset dates of filtered cases on a raster at a resolution of ~100 m. We then used a 5 × 5 cells sliding window to measure the slope of the interpolated surface to generate a friction map and smoothed the friction surface at an average 11 × 11 cells to prevent areas with no friction values[43]. From the resulting smoothed friction map, we estimated the wavefront velocity as the inverse of the friction values.

### Impact of population density, travel inaccessibility and international borders on the wavefront velocity

We assessed the impact of several covariates on the wavefront velocity of the two main cVDPV2 outbreaks—namely, human population density, travel inaccessibility[22] and international borders—by adapting a methodology used in landscape phylogeography to evaluate the impact of environmental factors on the dispersal dynamic of viral lineages[20,21]. To estimate travel inaccessibility, we used a 'friction surface'[22]—a map that quantifies resistance to human movement on the basis of landscape and infrastructure. Areas with high resistance take longer to cross, while areas with low resistance permit faster travel, with each grid cell presenting the time needed to travel 1 m[22]. This analytical procedure was previously implemented to investigate the environmental factors impacting the wavefront dynamic of the African swine fever outbreak that occurred in 2018–2019 in southern Belgium. It consisted

of a univariate regression approach to test if the heterogeneity of the dispersal velocity could be better explained when considering an environmental distance instead of a geographic distance. Specifically, we compared the amount of variability of the dispersal duration ($t$) that can be explained by an environmental distance ($d_{env}$), that is, a measure of spatial distance weighted according to the spatial heterogeneity of an underlying environmental layer, as opposed to the geographical distance ($d_{geo}$). In other words, we aimed to assess whether $d_{env}$ is more correlated with $t$ than $d_{geo}$ is. We did so by estimating and testing the level of significance of the metric $Q$, here defined as the proportion of heterogeneity in the wavefront dispersal velocity that can be associated with a given environmental factor[19], that is, the dispersal duration variability that could be explained when considering an underlying environmental heterogeneity. For a given outbreak, $Q$ was computed as the difference between two coefficients of determination: (1) $R^2_{geo}$ estimated from the linear regression between the dispersal durations and geographic distances ($t \sim d_{geo}$), where ~ means 'linear regression between', and (2) $R^2_{env}$ estimated from the linear regression between the dispersal durations and environmental distances ($t \sim d_{env}$). Both $d_{geo}$ and $d_{env}$ were computed between the outbreak origin (approximated by the location of the index case for the considered cVDPV2 outbreak) and the location of each filtered case (see above). Similarly, the corresponding dispersal durations $t$ were here obtained by the difference between the time of onset of each filtered case and the time of onset of the index case. As introduced above, focusing on filtered cases, that is, cases identified as contributing to the extension of the infected area, allows for the specific investigation of the wavefront dynamic of a given outbreak.

Both $d_{geo}$ and $d_{env}$ were computed with the path model implemented in the programme 'Circuitscape 5'[44,45], which uses circuit theory to treat rasters as conductance or resistance grids and compute environmental distances as pairwise electric resistances. While $d_{env}$ values were computed on the environmental rasters to test (rasters of population density, travel inaccessibility or international borders), $d_{geo}$ values were computed on a 'null' raster, that is, a uniform raster with the exact same resolution and extent but with a value of 1 assigned to all accessible raster cells (cells corresponding to maritime areas being assigned an 'NA' value). Distances computed with the Circuitscape 5 algorithm on such a null raster thus correspond to a proxy of geographic distances, that is, the cost of dispersal when not considering a particular environmental variable that might induce heterogeneity in the wavefront dispersal velocity. The uncertainty in the path taken is therefore also considered when computing $d_{geo}$, making this proxy for the geographic distance directly comparable to the environmental distances computed with the same algorithm on different environmental rasters.

The selection of a univariate approach is here motivated by the nature of the environmental distances computed with a path model such as the one implemented in Circuitscape 5. Because environmental distances computed with this algorithm remain correlated with the geographic distances, the resulting collinearity between environmental and geographic distances prevents the interest of relevant multivariate analyses that would attempt to predict the variability of the dispersal durations $t$ according to $d_{geo}$ and one or several environmental distances $d_{env}$ computed on distinct environmental rasters.

We tested different values for the rescaling parameter $k$ used to define the relationship between the original raster cell values and their conductance or resistance values. Specifically, for each environmental factor, we generated several distinct rasters by transforming the original raster cell values with the following formula: $v_t = 1 + k(v_o/v_{max})$, where $v_t$ and $v_o$ are the transformed and original cell values, and $v_{max}$ is the maximum cell value recorded in the raster. The rescaling parameter $k$ here allows the definition and testing of different strengths of raster cell conductance or resistance, relative to the conductance/resistance of a cell with a minimum value set to 1 (ref. 23). For each of the three environmental factors, we tested five different values for $k$ (that is, $k = 10, 100, 1,000, 10,000$ and $100,000$). While the friction raster was solely tested as a resistance factor, both the rasters for the log-transformed population density and international borders were tested as resistance and/or conductance factors in separate runs. As such, each run only tested one hypothesis, that is, the impact of a specific environmental variable acting either as a conductance or as a resistance factor on the wavefront velocity. In the main text, we only reported the results for the significant runs with the value of $k$ maximising the $Q$ statistic, that is, the value of $k$ that increases the correlation between the environmental distances and dispersal durations (see Supplementary Table 2 for the report of all $Q$ estimates).

To assess the statistical significance of the effect of each variable on the wavefront velocity, we performed a one-tailed hypothesis-testing by comparing the $Q$ obtained on the basis of the actual case dataset for each given outbreak to the distribution of $Q$ values obtained under a null dispersal model, that is, 100 null dispersal datasets obtained through a randomization procedure. Each null dispersal dataset was generated by randomly rotating the dispersal vectors around the outbreak origin, herein defined as the first case of the outbreak (earliest date of onset). This resulted in 100 distributions of cases, or outbreaks, that are similar to the observed geodesic distance and time duration from the outbreak origin, but that should not be associated with any environmental variable as their final location is decided at random. As such, the null hypothesis would be that the correlation between environmental distances and dispersal durations estimated for the actual case dataset is not higher than that estimated for the null dispersal model datasets. For the significant effects, we show both the estimated $Q$ for the actual outbreak dataset and the distribution of $Q$ values obtained under the null dispersal model.

## Type 2 population immunity estimates

OPV-induced monthly population immunity estimates against type 2 poliomyelitis in children aged 6–36 months per admin-2 were downloaded from the Polio Immunity Mapping page hosted within the Gates Foundation website and accessed through POLIS (pim.bmgf.io). The estimates were generated by the Institute of Disease Modelling, and the methods for generating such estimates have been described in detail elsewhere[18,46]. In brief, estimates are generated through a cohort model assuming children receive type 2 OPV either through routine immunization until the time of The Switch in April 2016 or through supplementary immunization activities (SIAs; mass campaigns) assuming either 50% or 80% per-campaign coverage. The age group 6–36 months was chosen, as this represents ~77% of all poliovirus cases. For routine immunization coverage, data on diphtheria–tetanus–pertussis is used, as polio vaccination is given at the same time as diphtheria–tetanus–pertussis. The immunity estimates do not consider immunity induced from the inactivated polio vaccine, as this does not induce a strong mucosal response in naive individuals and does not protect against transmission. The campaign calendar was downloaded from POLIS to obtain timing and geography of campaigns. Specific effectiveness is assumed for each vaccine type and can be found in the original publication. For our analysis, we only considered immunity estimates generated for a vaccination campaign coverage of 80%. Median and mean immunity estimates were calculated from the original estimates at the admin-2 level and are not weighted by population size, that is, for each country, median immunity represents the median value for the estimates for all admin-2 within that country for that specific month.

## Vaccination lag

We used AFP and SIA data downloaded from POLIS on 5 February 2024. Allowing 90 days for processing AFP cases, we excluded any cases with onset after 7 November 2023. In 11 of 19 countries with OPV SIAs and NIE-JIS-1 or NIE-ZAS-1 detections included in the analysis, geographical subdivisions have changed over the analysis period. We used the

current admin-2 divisions (ADM2) as our reference and assigned cases linked to old divisions by finding the current division that overlaps with a randomly assigned point within the old division. We assigned SIAs linked to old divisions to current divisions if at least 50% of the current division's area was covered by the old division. For each ADM2, we calculated the time from the first case of either NIE-JIS-1 or NIE-ZAS-1 in the ADM2 and the most recent type 2 OPV SIA response that came before the onset of this case.

### Public availability of poliovirus AFP VP1 sequences

All poliovirus genetic sequences and their associated metadata were downloaded from GenBank on 21 July 2022: 7,227 entries. Given the high frequency of GenBank entries missing important metadata (for example, collection date, country of collection, serotype and sample type) (Supplementary Table 4), we performed a thorough search of manuscripts and reports linked to all GenBank entries with missing data to fill in these gaps and reduce the number of sequences excluded from the dataset. We focused on VP1 sequences only, as this is the region routinely sequenced by the GPEI for virus identification and typing[47]; as such, sequences not including the VP1 capsid-protein region were removed from the dataset. After filtering for sequences containing the VP1 region (903 nucleotides) with a length of at least 800 bases (90% sequence coverage) that were not reference sequences, duplicates, missing country and/or date of collection or from animal/drug studies, we ended up with 5,057 entries from 98 countries. We further filtered our dataset by keeping only sequences that belong to a polio AFP case and that were typed to be either WPV or VDPV, that is, sequences belonging to environmental samples, contacts, healthy individuals and Sabin-like were removed from the dataset. Our final dataset contained 1,392 sequences from 33 countries covering the period between 2000 and 2020. An availability ratio, sequences over cases reported in the same period, was estimated using the number of AFP cases reported on POLIS, retrieving a global sequence availability of 7%. For availability estimates between 1980 and 2000, WPV and VDPV case counts were downloaded from the POLIS extranet[37]. The availability of VDPV and WPV AFP VP1 sequences is shown in Supplementary Figs. 11 and 13–15.

### Sequencing of additional historical VP1 WPV1 sequences

Additional VP1 sequences were obtained from historical WPV1 isolates available at the Medicines and Healthcare Products Regulatory Agency (MHRA), including viruses from 1953 to 2011 from all WHO regions and some that had been stored at the MHRA for >40 years. The requirement for institutional ethics approval and individual consent was waived, as sequences are generated as part of routine poliovirus surveillance from residual samples used for diagnostic purposes. While in this study we only had access to virus isolates and not human samples, diagnostic specimens were originally obtained through routine disease surveillance and polio eradication efforts. Polioviruses were isolated from clinical samples in primary kidney, MRC-5 or HEp-2c cells as part of local routine poliovirus surveillance and, after receiving the primary isolates from the MHRA, virus stocks were obtained by one passage in Hep-2c cells. All sample preparation for sequencing, genomic analyses and downstream data processing was carried out on anonymized material, identifiable solely by laboratory or epidemiological codes. All applicable ethical guidelines were strictly observed, as part of routine poliovirus surveillance carried out at the MHRA. Poliovirus sequences were obtained from reverse transcription polymerase chain reaction (RT–PCR) products by next generation sequencing using Illumina MiSeq and MinION nanopore platforms (ONT) as described before[48].

Viral RNA was extracted using the MagMax viral RNA extraction kit and reverse transcription was performed with one-step RT–PCR using a SuperScript III One-Step RT–PCR System with Platinum Taq High Fidelity DNA Polymerase (Invitrogen) and primers PCR-F

(5′-AGA GGC CCA CGT GGCGGC TAG −3′) and PCR-3′ (5′-CCG AAT TAA AGA AAA ATT TAC CCC TAC A −3′). Amplified poliovirus DNA was then sequenced by Illumina MiSeq and MinION nanopore. For Illumina MiSeq, the Nextera DNA Prep and Nextera DNA CD Indexes were used for library preparation and dual indexing. Samples were pooled together and 250 bp paired-end reads were generated using MiSeq v2 (500 cycles). bcl2fastq v2.2 (MiSeq) and DRAGEN BCL Convert v3.8.4 (NextSeq 2000) were used for demultiplexing of reads. FASTQ reads were trimmed and quality checked by Cutadapt v2.10 for a minimum Phred score of Q30, minimal read length of 75 bp, and zero ambiguous nucleotides. For ONT sequences, standard library preparation and barcoding were performed using SQK-NBD110-9/SQK-NBD112.96 and barcoding expansion EXP-PBC096 kits. NEBNext Ultra II End Repair/dA-Tailing Module (cat no. E7546L) was used for end-repair of PCR products, and Blunt/TA Ligase Master Mix (cat no. M0367) was used for barcoding. Purification of barcoded and pooled samples was performed using AmPure XP beads (cat no. A63882). Finally, NEBNext Quick Ligation Module (E6056L) was used for ligation of sequencing adaptors to the purified libraries, which were subsequently loaded onto primed R9.4 or R10.4 flow cells and run on an ONT MinION Mk1B. A high-accuracy basecalling model using Guppy v5 was used for basecalling of Fast5 files. FASTQ files from both Illumina and Nanopore sequencing were imported into Geneious 10.2.3, and consensus sequences were generated by reference-guided assembly and a minimum coverage of >200 throughout the genome[48].

### Collation and quality control of publicly available genetic sequences and metadata

Given the lack of sequences belonging to post-switch cVDPV2 outbreaks spanning multiple countries, we focused our phylogenetic analysis on publicly available WPV1 sequences to reconstruct the historical international spread of poliovirus in the regions currently affected by cVDPV2 outbreaks: Africa and South Asia. For quality control of the remaining genomes, sequences were visually inspected on AliView 1.28 (ref. 49), they were aligned using MAFFT v7.490 (ref. 50) and a maximum likelihood (ML) phylogeny was inferred using IQTREE 2 (ref. 51). The ML phylogeny was used for a root-to-tip regression of the genetic divergence against the collection date to identify sequences with potentially excessive mutations or wrong collection dates on TempEst v.1.5.3 (ref. 52). One sequence (HQ286311) was excluded owing to a high residual value—the only sequence with residual value exceeding ±0.2, which is indicative of a potential error in the reported collection date. As the correct collection date could not be confirmed, the sequence was removed from the dataset. The final dataset (dataset A) of global WPV1 VP1 sequences contained a total of 1,572 sequences, 1,534 publicly available sequences collected between 1958 and 2015 and 38 newly generated VP1 WPV1 sequences collected between 1953 and 2011. Discrete phylogeographic reconstruction (see below) was performed to identify clades of local transmission within Africa and South Asia. Clade 2 (201, dataset B) and clades 4a (304, dataset C) and 4b (559, dataset D) were further used to reconstruct poliovirus spread between countries in those regions. For further validation of our findings, we also analysed 123 WPV3 VP1 sequences (dataset E) with collection dates spanning 1980–2011. The same quality control pipeline applied for WPV1 and mentioned above was used for WPV3.

### Phylogenetic analysis

For all datasets (see 'Collation and quality control of publicly available genetic sequences and metadata'), an initial ML phylogeny was inferred using FastTree v.2.1 (ref. 53) under a GTR + I + G substitution model, and a final ML phylogeny was inferred using IQTREE v.2 under the best substitution model selected by ModelFinder[54] according to a Bayesian information criterion. VP1 sequences were screened for recombination using all method available in RDP4 (ref. 55), and no evidence of

recombination was found. Temporal signal was assessed on TempEst by root-to-tip regression of the genetic diversity against the date of collection with an $R^2$ of 0.58 for dataset A, 0.89 for dataset B, 0.96 for dataset C, 0.89 for dataset D and 0.64 for dataset E (Supplementary Fig. 20). Time-rooted phylogenies were inferred using BEAST v.1.10.5 (ref. [56]) under a GTR + I + G substitution model and an uncorrelated relaxed molecular clock for datasets A and B and a strict clock for datasets C–E. Molecular clock selection was based on the standard deviation and coefficient of variation for the uncorrelated lognormal relaxed clock rates for each dataset and whether they included/abuted zero or not (for details, see Supplementary Table 5). For the tree prior, we used a skygrid[57] model one grid per year from the most recent sampled sequence to the x-intercept in the root-to-tip regression. Bayesian analysis was run using BEAGLE v4.0.0 (ref. [58]) in four separate replicates for the following Markov chain Monte Carlo steps lengths. Convergence was assessed using Tracer v1.7.2 (ref. [59]). After removal of a 10% burn-in, posterior trees and log files were combined using LogCombiner 1.10.4 (ref. [60]), summarized into a maximum clade credibility phylogeny using TreeAnnotator 1.10.4 (ref. [60]) and visualized and edited using FigTree v1.4.4. A resampling of 1,000 trees was performed to be used as input for the discrete trait phylogeographic analysis (DTA) using LogCombiner 1.10.4 (see below). Details on specification used for each dataset are provided in Supplementary Table 6.

### Phylogeographic analysis of WPV1 VP1 sequences

We used an asymmetric DTA[61] approach implemented within BEAST v.1.10.5 to reconstruct the transitions between different geographic locations in the dataset and estimate the instantaneous movement rates between them. For the transition rate estimation, we used a Bayesian stochastic search variable selection[61] to find the most parsimonious diffusion process, which also allows for the estimation of a Bayes factor (BF) and identify the transition rates with statistical support. The number of transitions between locations along the tree was estimated on a branch-by-branch basis by using a Markov jumps approach[62]. To account for sampling bias, we adopted an adjusted BF approach, which relies on a tips-swapping procedure for the locations associated with each tip on the tree and estimate the posterior probability for all possible transition rates on the basis of the swapped location states at each iteration[63]. This new posterior probability can then be used as the new baseline to estimate an adjusted BF for the analysis without the tips-swapping approach and to which only the actual location associated with each tip was considered to estimate the transition rates[64]. By performing this correction, we could increase our confidence that the observed significant rates are a result of phylogenetic signal, rather than sampling bias. We then presented only rates with an adjusted BF support of at least 10, which is more strict than the conventional threshold of >3 that is normally used (positive statistical support as defined by Kass and Raftery[65]). The same approach was used to confirm Nigeria as the root location for the clade 2 subtree, retrieving a BF support of 19.

We considered four different schemes of discretization, one for the global analysis and one for each of the three clades of study: (1) for the global phylogeny (dataset A), sequences were discretized in 13 world regions (Central Africa, Central America and the Caribbean, Central Asia, East Asia, Eastern Africa, Europe, the Middle East, North America, Northern Africa, South America, South Asia, Southeast Asia and Western Africa); (2) for clade 2 analysis (dataset B), sequences were discretized according to country of collection ($n$ = 22; Angola, Benin, Brazil, Burkina Faso, Central African Republic, Cameroon, Cape Verde, Chad, Cote d'Ivoire, the DRC, Egypt, Equatorial Guinea, Ethiopia, Ghana, Guinea, Mauritania, Niger, Nigeria, Sierra Leone, Sudan and Zambia); (3) for clade 4a analysis (dataset C), sequences were discretized according to country of collection ($n$ = 9; Afghanistan, Bulgaria, China, Egypt, India, Israel, Nepal, Pakistan and Syria); and (4) for clade 4b, sequences were also discretized according to country of collection

($n$ = 11; Angola, Congo, the DRC, Egypt, India, Kazakhstan, Namibia, Nepal, Russia, Tajikistan and Turkmenistan). To reduce computational time, DTAs for all datasets were run on 1,000 resampled posterior trees from the standard BEAST analysis[33,66] (see 'Phylogenetic analysis'). For each dataset, four independent BEAST runs were performed for a length of 10 million states. Log files and trees were assessed and summarized as aforementioned.

Phylogenetic clades on the main global dataset (dataset A) were defined according to the DTA posterior location support and the bootstrapping (ML tree) and posterior (BEAST analysis) support for the internal nodes. We considered only supports ≥0.9/90%. By doing so and given the regions of interest for the epidemiology of cVDPV2, we identified two main clades of interest: clades 2 and 4. Sequences in clade 2 represent mostly the process of spread between and within Central and Western Africa, while sequences in clade 4 are associated with spread in South Asia, Middle East, Central Africa and Central Asia. Clade 4 was further subdivided into clades 4a and 4b, as they clearly represent two very separate spread processes; while clade 4a contains sequences from South Asia and the Middle East, clade 4b contains sequences from South Asia, Central Asia and Central Africa. Clades 4a and 4b were separated at their most recent common ancestor with posterior node and location (South Asia) supports equal to 1.

### Reporting summary

Further information on research design is available in the Nature Portfolio Reporting Summary linked to this article.

## Data availability

Genetic sequences are available at GenBank under the following accession numbers: PQ159203–PQ159240. Detailed disease surveillance data on which this research is based are available from the Word Health Organization (WHO) Institutional Data Access/Ethics Committee for Global Polio Eradication Initiative research partners who meet the criteria for access to confidential data. Epidemiological and immunity data analysed in this study were obtained from the WHO Polio Information System. These data are the property of the individual countries, and data access was provided through the Global Polio Eradication Initiative (GPEI) data sharing agreement. Data are available from the WHO Institutional Data Access/Ethics Committee for GPEI research partners who meet the criteria for access to confidential data (https://extranet.who.int/polis/).

## Code availability

All code used for the wavefront velocity data analysis is available via GitHub at https://github.com/sdellicour/vdpv_wavefront.

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

## Acknowledgements

We thank all polio surveillance officers for their essential contributions to the successful implementation of this study. The publication of the maps in this manuscript does not imply the expression of any opinion whatsoever on the part of the World Health Organization (WHO) concerning the legal status of any territory, city or area or of its authorities, or concerning the delimitation of its frontiers or boundaries. This publication is based on research funded in whole or in part by the Gates Foundation. The findings and conclusions contained within are those of the authors and do not necessarily reflect the official positions or policies of the Gates Foundation. We acknowledge the funding provided by the Gates Foundation (grant nos. INV-031605 to I.M.B. and N.C.G. and INV-043859 to J.M.), the Fonds National de la Recherche Scientifique (FNRS, Belgium; grant no. F.4515.22 to S.D.), the Research Foundation—Flanders (Fonds voor Wetenschappelijk Onderzoek—Vlaanderen; grant no. G098321N to S.D.), the European Union Horizon 2020 project MOOD (grant no. 874850 to S.D.) and the São Paulo Research Foundation (FAPESP; grant nos. 2024/10682-6, 2022/15985-1 and 2019/21858-0 to C.A.P.). We thank A. L. van Wezel (National Institute of Public Health, Bilthoven, The Netherlands), R. Nájera (Instituto de Salud Carlos III, Madrid, Spain), H. G. Pereira (Instituto Oswaldo Cruz, Rio de Janeiro, Brazil/ National Institute for Medical Research (NIMR), London, UK), J. H. Nakano (Centers for Disease Control and Prevention, Atlanta, USA), H. Asghar (National Institute of Health, Islamabad, Pakistan/WHO EMRO), B. D. Schoub (National Institute for Communicable Diseases, Johannesburg, South Africa), M. Yin-Murphy (National University of Singapore), F. Horaud (Institut Pasteur, Paris, France), I. M. Forgie (MRC Laboratories, Fajara, the Gambia), B. Le Guenno (Institut Pasteur de Dakar, Senegal), J. K. Odoom (Noguchi Memorial Institute for Medical Research, University of Ghana), D. Carrington (Glasgow Royal Infirmary, University of Glasgow, UK) and S. Nejmi (National Institute of Hygiene/Institut Pasteur du Maroc, Rabat, Morocco) for providing poliovirus isolates sequenced in this study. We also thank R. Mate and P. Sinnakandu from the Medicines and Healthcare products Regulatory Agency (MHRA) for support with Illumina sequencing and all surveillance and laboratory teams who have been fundamental for the collection and processing of polio surveillance data used in this study.

## Author contributions

Conception: D.d.S.C., S.D., N.C.G. and I.M.B.; Data curation: D.d.S.C., L.V.C., D.J., A.V., H.L., J.M. and I.M.B.; Formal analysis: D.d.S.C., S.D., L.V.C., C.A.P., C.B.U. and I.M.B.; Investigation: D.d.S.C., S.D., D.K., M.M., K.A. and J.M.; Methodology: D.d.S.C., S.D. and I.M.B.; Project administration: D.d.S.C. and I.M.B.; Resources: S.D., J.M., N.C.G. and I.M.B.; Software: D.d.S.C., S.D. and L.V.C.; Supervision: D.d.S.C., N.C.G. and I.M.B.; Validation: D.d.S.C. and S.D.; Visualization: D.d.S.C. and S.D.; Writing—original draft: D.d.S.C. and I.M.B.; Writing—review and editing: D.d.S.C., S.D., L.V.C., C.A.P., D.J., C.B.U., A.V., H.L., D.K., M.M., K.A., C.M.P., A.S.B., J.M., N.C.G. and I.M.B.; Funding acquisition: C.M.P., A.S.B., J.M., N.C.G. and I.M.B.

## Competing interests

A.S.B., C.M.P., A.V. and H.L. are employees of the Gates Foundation, which funded this work. I.M.B. reports grants to her institution from the World Health Organization and the Gates Foundation. N.C.G. reports grants to his institution from the UK Medical Research Council, the World Health Organization and the Gates Foundation. J.V. reports funding from the Gates Foundation. The funders had no role in the study design, data collection and analysis, decision to publish or preparation of the manuscript. The other authors declare no competing interests.

## Additional information

**Correspondence and requests for materials** should be addressed to Darlan da Silva Candido.

# Reporting Summary

## Statistics

For all statistical analyses, confirm that the following items are present in the figure legend, table legend, main text, or Methods section.

| n/a | Confirmed | |
|---|---|---|
| ☐ | ☒ | The exact sample size (*n*) for each experimental group/condition, given as a discrete number and unit of measurement |
| ☒ | ☐ | A statement on whether measurements were taken from distinct samples or whether the same sample was measured repeatedly |
| ☐ | ☒ | The statistical test(s) used AND whether they are one- or two-sided<br>*Only common tests should be described solely by name; describe more complex techniques in the Methods section.* |
| ☐ | ☒ | A description of all covariates tested |
| ☐ | ☒ | A description of any assumptions or corrections, such as tests of normality and adjustment for multiple comparisons |
| ☐ | ☒ | A full description of the statistical parameters including central tendency (e.g. means) or other basic estimates (e.g. regression coefficient) AND variation (e.g. standard deviation) or associated estimates of uncertainty (e.g. confidence intervals) |
| ☒ | ☐ | For null hypothesis testing, the test statistic (e.g. *F*, *t*, *r*) with confidence intervals, effect sizes, degrees of freedom and *P* value noted<br>*Give P values as exact values whenever suitable.* |
| ☐ | ☒ | For Bayesian analysis, information on the choice of priors and Markov chain Monte Carlo settings |
| ☒ | ☐ | For hierarchical and complex designs, identification of the appropriate level for tests and full reporting of outcomes |
| ☒ | ☐ | Estimates of effect sizes (e.g. Cohen's *d*, Pearson's *r*), indicating how they were calculated |

*Our web collection on statistics for biologists contains articles on many of the points above.*

## Software and code

Policy information about availability of computer code

| Data collection | Epidemiological data was downloaded from POLIS. |
|---|---|
| Data analysis | Data visualization was performed using ggplot2 in R. Wavefront velocity analysis was performed using a custom made code developed by Simon Dellicour based on the use of circuitscape 5 in R (the code has been made available on github: https://github.com/sdellicour/vdpv_wavefront). For sequencing and phylogenetic analysis, all softwares used are available for free: Guppy v5, Geneious 10.2.3, BEAST v.1.10.5, BEAGLE v4.0.0 , IQTREE v.2 , FastTree v.2.1, Tracer v1.7.2, FigTree v1.4.4, TempEst v.1.5.3, TreeAnnotator 1.10.4 , LogCombiner 1.10.4 60, and AliView 1.28. |

For manuscripts utilizing custom algorithms or software that are central to the research but not yet described in published literature, software must be made available to editors and reviewers. We strongly encourage code deposition in a community repository (e.g. GitHub). See the Nature Portfolio guidelines for submitting code & software for further information.

# Data

Policy information about availability of data

All manuscripts must include a data availability statement. This statement should provide the following information, where applicable:
- Accession codes, unique identifiers, or web links for publicly available datasets
- A description of any restrictions on data availability
- For clinical datasets or third party data, please ensure that the statement adheres to our policy

> Genetic sequences are available on GenBank under the following accession numbers: PQ159203-PQ159240. Detailed disease surveillance data on which this research is based are available from the WHO Institutional Data Access/Ethics Committee for Global Polio Eradication Initiative research partners who meet the criteria for access to confidential data.

# Research involving human participants, their data, or biological material

Policy information about studies with human participants or human data. See also policy information about sex, gender (identity/presentation), and sexual orientation and race, ethnicity and racism.

| | |
|---|---|
| Reporting on sex and gender | Our study focuses on the spatiotemporal patterns of poliovirus spread rather than on demographic and clinical characteristics of infected patients and only uses surveillance data from the WHO POLIS.Epidemiological data on all acute flaccid paralysis(AFP) cases with paralysis onset after "The Switch", May 1, 2016, through September 29, 2023, was downloaded from the Polio Information System (POLIS). Sequences were obtained from historical WPV1 isolates available at Medicines and Healthcare products Regulatory Agency (MHRA), including viruses from 1953 to 2011 from all WHO regions and some which had been stored at MHRA for more than 40 years and demographic information is not available. |
| Reporting on race, ethnicity, or other socially relevant groupings | Please see above. |
| Population characteristics | Please see above. |
| Recruitment | No recruitment was performed as this is a study based on surveillance data. Epidemiological data on all acute flaccid paralysis(AFP) cases with paralysis onset after "The Switch", May 1, 2016, through September 29, 2023, was downloaded from the Polio Information System (POLIS). For the epidemiological data, potential biases are related to differences in the surveillance capacity and reporting across different countries. Sequences were obtained from historical WPV1 isolates available at Medicines and Healthcare products Regulatory Agency (MHRA), including viruses from 1953 to 2011 from all WHO regions and some which had been stored at MHRA for more than 40 years. |
| Ethics oversight | Imperial College Research Governance and Integrity Team (reference ID 21IC6996) |

Note that full information on the approval of the study protocol must also be provided in the manuscript.

# Field-specific reporting

Please select the one below that is the best fit for your research. If you are not sure, read the appropriate sections before making your selection.

☒ Life sciences    ☐ Behavioural & social sciences    ☐ Ecological, evolutionary & environmental sciences

For a reference copy of the document with all sections, see nature.com/documents/nr-reporting-summary-flat.pdf

# Life sciences study design

All studies must disclose on these points even when the disclosure is negative.

| | |
|---|---|
| Sample size | Study based on surveillance data. All available surveillance data was used for analysis and, as such, no sample size calculation was performed. Epidemiological data on all acute flaccid paralysis (AFP) cases with paralysis onset after "The Switch", May 1, 2016, through September 29, 2023, were downloaded from the Polio Information System (POLIS),. Briefly, the polio surveillance system involves the surveillance of AFP cases, environmental samples, contact sampling and community sampling. Poliomyelitis cases are laboratory confirmed according to WHO guidelines using virus isolation in cell culture, intertypic differentiation and genetic sequencing of the VP1 capsid protein. |
| Data exclusions | No exclusions were performed. All available epidemiological and genomic data was used meeting the minimum criteria specified in methods was used. For genomic data, only sequences including the VP1 region of WPV1 poliovirus, used routinely for surveillance, and for which data on country and collection data were available were used. Please see methods. |
| Replication | Study based on surveillance data. Not applicable. |
| Randomization | Study based on surveillance data. Not applicable. |
| Blinding | Study based on surveillance data. Not applicable. |

# Reporting for specific materials, systems and methods

We require information from authors about some types of materials, experimental systems and methods used in many studies. Here, indicate whether each material, system or method listed is relevant to your study. If you are not sure if a list item applies to your research, read the appropriate section before selecting a response.

## Materials & experimental systems

| n/a | Involved in the study |
|-----|----------------------|
| ☒ ☐ | Antibodies |
| ☒ ☐ | Eukaryotic cell lines |
| ☒ ☐ | Palaeontology and archaeology |
| ☒ ☐ | Animals and other organisms |
| ☒ ☐ | Clinical data |
| ☒ ☐ | Dual use research of concern |
| ☒ ☐ | Plants |

## Methods

| n/a | Involved in the study |
|-----|----------------------|
| ☒ ☐ | ChIP-seq |
| ☒ ☐ | Flow cytometry |
| ☒ ☐ | MRI-based neuroimaging |

## Plants

| | |
|---|---|
| Seed stocks | *Report on the source of all seed stocks or other plant material used. If applicable, state the seed stock centre and catalogue number. If plant specimens were collected from the field, describe the collection location, date and sampling procedures.* |
| Novel plant genotypes | *Describe the methods by which all novel plant genotypes were produced. This includes those generated by transgenic approaches, gene editing, chemical/radiation-based mutagenesis and hybridization. For transgenic lines, describe the transformation method, the number of independent lines analyzed and the generation upon which experiments were performed. For gene-edited lines, describe the editor used, the endogenous sequence targeted for editing, the targeting guide RNA sequence (if applicable) and how the editor was applied.* |
| Authentication | *Describe any authentication procedures for each seed stock used or novel genotype generated. Describe any experiments used to assess the effect of a mutation and, where applicable, how potential secondary effects (e.g. second site T-DNA insertions, mosiacism, off-target gene editing) were examined.* |

