## [Peer Review File · Nature Microbiology]

Historical and current spatiotemporal patterns of wild and vaccine-derived poliovirus spread

Corresponding Author: Dr Darlan Candido

Version 0:

Reviewer comments:

Reviewer #1

(Remarks to the Author)

The authors define the spatiotemporal patterns of spread for VDPV2 and WT1 poliovirus. The former is defined using available case data collected during surveillance for AFP cases, which includes locations and time. The latter is based on whole genome sequences. The major contribution is quantifying the rate of spread, primarily in several large outbreaks in Africa. These data can theoretically be used to better time and target vaccination campaigns to control outbreaks. More speculative, and less substantiated, is an analysis relating cross border spread to the levels of immunity. This is interesting, but - as the authors indicate - an area that needs more investigation. With respect to their data on spread across borders, of the 76 outbreaks, only 8 spread to other countries.

Overall, this builds on prior work looking at factors influencing emergence and spread of VDPV, a major issue in the polio eradication campaign. These prior studies are well cited and described. For example, one that link emergences of VDPV as associated with large campaigns in Nigeria, Ethiopia, and DRC in populations with low immunity and no follow up campaigns.

This reviewer found that Q statistic a little hard to grasp. Given that it is important for the inference, suggest that this be specifically reviewed by others.

Major critiques

I am struggling to interpret the differences between Figs 3B and S11. The latter includes outliers and many of the outbreaks have extremely long tails of rates that appear to have large number of wavefront estimates (dots). With this level of skew - with many wavefront estimates of 100-400 km/day, what is one to think of the conclusions regarding differences in the 1-10 km/day range? Yes there are fast movements involved, but what does this mean for inferences regarding the low rates and the importance of the factors involved? Similarly, the authors claim that the wavefront can be used to target and time vaccination campaigns. Would one to that based on the median or the tail of the skew?

The main "sell" of the manuscript - the relationship between immunity and the impact of borders on rate of spread is based on comparison of two outbreaks with opposing impacts of borders. This seems tenuous.

I agree with the authors that it is a major limitation that there weren't sequences available for VDPV2. These should exist, as the authors indicate. Why are they not available for inclusion in the analysis. This would seem to be important to verify that the outbreaks are ascertained correctly with respect to emergences and sources.

Minor critiques

Figure S8 is mislabeled S10 in supplement.

Figure S10 (the real one) - isn't entirely convincing as displayed in terms of relating speed of spread to the levels of immunity. For example, NIE-SOS-7 and ANG-HUI-1 have different color patterns at the levers (ends of regression). And also looking at NIE-JIS-1, supports lines 149-152 in text.

Lines 248-250 Is it really striking that most spread happens between neighboring countries?

(Remarks on code availability)

Reviewer #2

(Remarks to the Author)

The manuscript investigates the spatiotemporal spread of poliovirus, focusing on differences between wild poliovirus type 1 (WPV1) and circulating vaccine-derived poliovirus type 2 (cVDPV2). Using wavefront velocity models, phylogeographic analysis, and epidemiological data, this study examines factors influencing outbreak size and speed of transmission. The findings highlight the role of vaccination coverage, population immunity, and geographic barriers in shaping virus dispersal patterns.

This study is overall of high quality and provides important insights for poliovirus surveillance and outbreak response strategies, with implications for global eradication efforts. The methodology is robust, comprehensive and in parts innovative. It is clearly described and provides sufficient level of detail, with maybe a couple of exceptions (see minor comments below). The results are also clearly reported and the choice of figures adequate. The discussion is strong. If anything, it could maybe better highlight policy implications in this section. It would be useful, for instance, to indicate in more detail how these results are translatable and who policymakers could act on the findings of this study.

Minor comments

1. Page 5, Figure 2: The colour scheme used in this figure makes the dots difficult to distinguish, especially in panels A and B. It is also difficult to distinguish the different shapes in panel D, due to their small size. The authors should consider revising this.
2. Page 6, Figure 3, panel A: on the Y axis, 'outbrak' should be 'outbreak'.
3. Page 6, Figure 3, panel D: should 'Côte d'Ivoire' be 'Ivory Coast'? Or, if the countries are listed in their official language, should 'Guinea' be 'Guinée'?
4. Line 536: More detail on the method used to identify sequences with potentially excessive mutations or wrong collection dates on TempEst v.1.5.3 should be provided.
5. Page 9, Figure 4, panel A: the branch supports / posterior probabilities of the branches should be shown on the phylogenies, one way or another. I appreciate this may make the trees less readable, but it is necessary for the interpretation of the phylogenies. I would also increase the size of panel A relative to panel B (the map is useful but probably less informative than the tree itself).
6. Line 554: The authors have adopted an uncorrelated relaxed molecular clock for all analyses. It would be useful to justify this choice of model, as it is likely to influence the inference of temporal patterns in this study. Was a strict molecular clock also tested against the datasets? If so, how?

(Remarks on code availability)

I do not have the expertise required to review the code provided with the manuscript.

Reviewer #3

(Remarks to the Author)

In "Unravelling spatiotemporal heterogeneities of wild and vaccine-derived poliovirus spread: past and present" the authors use cases of cVDPV2 (vaccine-derived cases of wild-type poliovirus 2) and sequences of WPV1 (wild-type poliovirus 1) to explore spatiotemporal patterns and drivers of polioviruses. The paper's primarily focus is on cVDPV2 outbreaks, given the importance for recommended vaccination programs, current global spread, and geographically widespread outbreak risk. Overall, the authors conduct a thorough analysis of both cVDPV2 and WPV1 data, however the relationship between cVDPV2 and WPV1 outbreaks should be strengthened.

Major comments:

For each outbreak, is it possible to provide a table with the name (abbreviation), time the outbreak began (possibly just a month-year or year), what other countries this outbreak spread to, aggregated case counts for the countries, and duration of outbreak. I recognize that the authors state there are limitations to sharing the line list data, however, without additional information it is difficult to recreate the results presented or interpret them in context.

The WPV1 analysis is sound on its own, however the authors do not provide support that cVDPV2 would spread like WPV1 and hence that the WPV1 phylogenetic results would be relevant. Since there are cases and sequences in several of the countries they consider, a more direct comparison, i.e. showing that results from the analysis of cases of WPV1 are like the results identified for cases of cVDPV2 would provide a clearer link. For example, are the outbreaks of cVDPV2 and WPV1 of similar size? Geographic spread? Duration? Or, although there may not be sufficient samples, an analysis of the cVDPV2 sequences should be done. Even if the phylogenetic analysis of cVDPV2 sequences is quite limited, it would be helpful to bridge the gap between the inference about WPV1 outbreaks and cVDPV2 outbreaks. As an aside, the figures on the WPV1 outbreak analysis are great – particularly Figures S15-S17.

The issue of immunity is a bit unclear if these are considered in the models directly, or used to explain deviations from the predicted model after the initial model is fit.

The authors exclude cases that do not extend the outbreak wavefront in their analysis of cVDPV2 cases. However, wouldn't this

bias the outbreak velocity measures? For example, if there was an outbreak with many cases, but these were all geographically localized, i.e. within the same location, then this would have an outbreak velocity of 0. This seems to be important to consider, since this would provide a clearer picture of all the cVDP2 outbreaks. Further, excluding additional local cases removes information about the true spatial spread of the outbreak. For example, if there were several cases all local and then on day n there was an outbreak much further away, then this would be relevant information since it would show that there are differences in the outbreak velocity over the duration of the outbreak.

The authors primarily rely on physical distance between locations, however have the authors considered other measures such as travel time distance – of which there are publicly available measures – to help consider the local geography and road network?

The authors only consider reported cases of cVDP2, which is known for being underreported. Some analysis to consider the issues of sampling, particularly geographic biases, should be conducted.

[Added after correspondence with editor]: The authors calculate a Q statistic, that measures the difference in R2 values from a regression using environmental variables or one that only takes into account distance. This metric has been used in multiple phylogeographic analyses to explore the extent to which environmental distances (that take into account various spatial environmental rasters) over a null model. While this metric has been used in multiple peer reviewed manuscripts, (roughly 100 papers have cited the original Dellicour et al paper), it is not clear why this statistic is used instead of conducting a proper statistical model comparison. For example, it is well known that R2 increases with the addition of any explanatory variable (junk or otherwise), regardless of their importance or relationship with the outcome of interest. Further, for model comparisons, it is more common to use something like a likelihood ratio test, as a formal test, as opposed to comparing the summary measures or at minimum a summary statistic that accounts for the possibility of introducing unnecessary and unrelated predictors. If a full likelihood could not be constructed, then quasilielihood approaches could also be considered. Finally, when conducting model selection, maximizing the difference in R2 is not a robust way to choose which variables are the most scientifically relevant or predictive of to explain the variability in the outcome of interest.

Minor comments:

While the outbreak abbreviation names may be familiar to the authors, or those who work primarily in polio, to a general reader the naming conventions are not clear and difficult to follow. Is it possible to also provide the full names of these outbreaks (and use the abbreviated names for figures, tables, etc.)?

Figure 1 part C – could more distinct colors be used for the different regions to make it easier to tell which outbreaks had broader geographic expansion than others?

The authors state that, outbreaks have emerged in 19 different countries and spread to at least 22 countries, however you cannot tell this from the figure since the many points are clustered together and overlapping. Further, you cannot tell which outbreaks are associated with different countries from this plot. Additional figures that are clearer would be helpful to the reader.

How was the threshold of only considering outbreaks with 20 or more cases determined? Particularly since polio is highly underreported, do the authors have a sense of what estimated outbreak sizes they are excluding with this threshold? Or if the results hold with a lower threshold?

Figures S9 and S10, S12 are all incredibly difficult to read and should be expanded to make it easier to understand the results.

(Remarks on code availability)

For double blind reviews, I recommend not hosting the code on GitHub since that is often associated with a (very identifiable GitHub user). I could not run the code, because a number of files were missing, for example - Human_popD.tif, Outbreak_results/NA.csv. Further, no code is provided (or a workflow) to conduct the phylogenetic analysis on the GitHub link.

Decision Letter:

24th March 2025

Dear Darlan,

Thank you for your patience while your manuscript "Unravelling spatiotemporal heterogeneities of wild and vaccine-derived poliovirus spread: past and present" was under peer-review at Nature Microbiology. Your manuscript has been seen by 3 referees, whose expertise and comments you will find at the end of this email. You have seen two out of three reports before and kindly provided a rebuttal to address the concerns raised by referee #1 and #3. You saw that although the reviewers find your work of some potential interest, they have raised a number of concerns that will need to be addressed before we can consider publication of the work in Nature Microbiology.

We would in general agree with your revision plan. In addition, we would kindly ask you to address the points raised by referee #2, as well as the concerns about the use of Q statistics mentioned by referee #3. The last point was mentioned after a correspondence with referee #3, as we needed to check whether this aspect has been evaluated or not. The added comment

has been clearly marked.

We would send a suitably revised manuscript back to the original referees for further evaluation.

Please include a data availability statement as a separate section after Methods but before references, under the heading "Data Availability". This section should inform readers about the availability of the data used to support the conclusions of your study. This information includes accession codes to public repositories (data banks for protein, DNA or RNA sequences, microarray, proteomics data etc...), references to source data published alongside the paper, unique identifiers such as URLs to data repository entries, or data set DOIs, and any other statement about data availability. At a minimum, you should include the following statement: "The data that support the findings of this study are available from the corresponding author upon request", mentioning any restrictions on availability. If DOIs are provided, we also strongly encourage including these in the Reference list (authors, title, publisher (repository name), identifier, year). For more guidance on how to write this section please see: <http://www.nature.com/authors/policies/data/data-availability-statements-data-citations.pdf>

* If you have not done so already we suggest that you begin to revise your manuscript so that it conforms to our Article format instructions at <http://www.nature.com/nmicrobiol/info/final-submission>. Refer also to any guidelines provided in this letter.

When submitting the revised version of your manuscript, please pay close attention to our [href="https://www.nature.com/nature-portfolio/editorial-policies/image-integrity">Digital Image Integrity Guidelines.](https://www.nature.com/nature-portfolio/editorial-policies/image-integrity) and to the following points below:

EXTENDED DATA FIGURES

Link Redacted

Note: This url links to your confidential homepage and associated information about manuscripts you may have submitted or be reviewing for us. If you wish to forward this e-mail to co-authors, please delete this link to your homepage first.

Nature Microbiology is committed to improving transparency in authorship. As part of our efforts in this direction, we are now requesting that all authors identified as 'corresponding author' on published papers create and link their Open Researcher and Contributor Identifier (ORCID) with their account on the Manuscript Tracking System (MTS), prior to acceptance. This applies to primary research papers only. ORCID helps the scientific community achieve unambiguous attribution of all scholarly contributions. You can create and link your ORCID from the home page of the MTS by clicking on 'Modify my Springer Nature account'. For more information please visit www.springernature.com/orcid.

If you wish to submit a suitably revised manuscript we would hope to receive it within 6 months. If you cannot send it within this time, please let us know. We will be happy to consider your revision, even if a similar study has been accepted for publication at Nature Microbiology or published elsewhere (up to a maximum of 6 months).

Yours sincerely,

Reviewer Expertise:

Referee #1: Virus evolution, infectious disease

Referee #2: Epidemiology modeling, infectious disease

Referee #3: Epidemiology, virology, modeling

Reviewer Comments:

Reviewer #1 (Remarks to the Author):

The authors define the spatiotemporal patterns of spread for VDPV2 and WT1 poliovirus. The former is defined using available case data collected during surveillance for AFP cases, which includes locations and time. The latter is based on whole genome sequences. The major contribution is quantifying the rate of spread, primarily in several large outbreaks in Africa. These data can theoretically be used to better time and target vaccination campaigns to control outbreaks. More speculative, and less substantiated, is an analysis relating cross border spread to the levels of immunity. This is interesting, but - as the authors indicate - an area that needs more investigation. With respect to their data on spread across borders, of the 76 outbreaks, only 8 spread to other countries.

Overall, this builds on prior work looking at factors influencing emergence and spread of VDPV, a major issue in the polio eradication campaign. These prior studies are well cited and described. For example, one that link emergences of VDPV as associated with large campaigns in Nigeria, Ethiopia, and DRC in populations with low immunity and no follow up campaigns.

This reviewer found that Q statistic a little hard to grasp. Given that it is important for the inference, suggest that this be specifically reviewed by others.

Major critiques

I am struggling to interpret the differences between Figs 3B and S11. The latter includes outliers and many of the outbreaks have extremely long tails of rates that appear to have large number of wavefront estimates (dots). With this level of skew - with many wavefront estimates of 100-400 km/day, what is one to think of the conclusions regarding differences in the 1-10 km/day range? Yes there are fast movements involved, but what does this mean for inferences regarding the low rates and the importance of the factors involved? Similarly, the authors claim that the wavefront can be used to target and time vaccination campaigns. Would one to that based on the median or the tail of the skew?

The main "sell" of the manuscript - the relationship between immunity and the impact of borders on rate of spread is based on comparison of two outbreaks with opposing impacts of borders. This seems tenuous.

I agree with the authors that it is a major limitation that there weren't sequences available for VDPV2. These should exist, as the authors indicate. Why are they not available for inclusion in the analysis. This would seem to be important to verify that the outbreaks are ascertained correctly with respect to emergences and sources.

Minor critiques

Figure S8 is mislabeled S10 in supplement.

Figure S10 (the real one) - isn't entirely convincing as displayed in terms of relating speed of spread to the levels of immunity. For example, NIE-SOS-7 and ANG-HUI-1 have different color patterns at the levers (ends of regression). And also looking at NIE-JIS-1, supports lines 149-152 in text.

Lines 248-250 Is it really striking that most spread happens between neighboring countries?

Reviewer #2 (Remarks to the Author):

The manuscript investigates the spatiotemporal spread of poliovirus, focusing on differences between wild poliovirus type 1 (WPV1) and circulating vaccine-derived poliovirus type 2 (cVDPV2). Using wavefront velocity models, phylogeographic analysis, and epidemiological data, this study examines factors influencing outbreak size and speed of transmission. The findings highlight the role of vaccination coverage, population immunity, and geographic barriers in shaping virus dispersal patterns.

This study is overall of high quality and provides important insights for poliovirus surveillance and outbreak response strategies, with implications for global eradication efforts. The methodology is robust, comprehensive and in parts innovative. It is clearly described and provides sufficient level of detail, with maybe a couple of exceptions (see minor comments below). The results are also clearly reported and the choice of figures adequate. The discussion is strong. If anything, it could maybe better highlight

policy implications in this section. It would be useful, for instance, to indicate in more detail how these results are translatable and who policymakers could act on the findings of this study.

Minor comments

1. Page 5, Figure 2: The colour scheme used in this figure makes the dots difficult to distinguish, especially in panels A and B. It is also difficult to distinguish the different shapes in panel D, due to their small size. The authors should consider revising this.
2. Page 6, Figure 3, panel A: on the Y axis, 'outbrak' should be 'outbreak'.
3. Page 6, Figure 3, panel D: should 'Côte d'Ivoire' be 'Ivory Coast'? Or, if the countries are listed in their official language, should 'Guinea' be 'Guinée'?
4. Line 536: More detail on the method used to identify sequences with potentially excessive mutations or wrong collection dates on TempEst v.1.5.3 should be provided.
5. Page 9, Figure 4, panel A: the branch supports / posterior probabilities of the branches should be shown on the phylogenies, one way or another. I appreciate this may make the trees less readable, but it is necessary for the interpretation of the phylogenies. I would also increase the size of panel A relative to panel B (the map is useful but probably less informative than the tree itself).
6. Line 554: The authors have adopted an uncorrelated relaxed molecular clock for all analyses. It would be useful to justify this choice of model, as it is likely to influence the inference of temporal patterns in this study. Was a strict molecular clock also tested against the datasets? If so, how?

Reviewer #2 (Remarks on code availability):

I do not have the expertise required to review the code provided with the manuscript.

Reviewer #3 (Remarks to the Author):

In "Unravelling spatiotemporal heterogeneities of wild and vaccine-derived poliovirus spread: past and present" the authors use cases of cVDPV2 (vaccine-derived cases of wild-type poliovirus 2) and sequences of WPV1 (wild-type poliovirus 1) to explore spatiotemporal patterns and drivers of polioviruses. The paper's primarily focus is on cVDPV2 outbreaks, given the importance for recommended vaccination programs, current global spread, and geographically widespread outbreak risk. Overall, the authors conduct a thorough analysis of both cVDPV2 and WPV1 data, however the relationship between cVDPV2 and WPV1 outbreaks should be strengthened.

Major comments:

For each outbreak, is it possible to provide a table with the name (abbreviation), time the outbreak began (possibly just a month-year or year), what other countries this outbreak spread to, aggregated case counts for the countries, and duration of outbreak. I recognize that the authors state there are limitations to sharing the line list data, however, without additional information it is difficult to recreate the results presented or interpret them in context.

The WPV1 analysis is sound on its own, however the authors do not provide support that cVDPV2 would spread like WPV1 and hence that the WPV1 phylogenetic results would be relevant. Since there are cases and sequences in several of the countries they consider, a more direct comparison, i.e. showing that results from the analysis of cases of WPV1 are like the results identified for cases of cVDPV2 would provide a clearer link. For example, are the outbreaks of cVDPV2 and WPV1 of similar size? Geographic spread? Duration? Or, although there may not be sufficient samples, an analysis of the cVDPV2 sequences should be done. Even if the phylogenetic analysis of cVDPV2 sequences is quite limited, it would be helpful to bridge the gap between the inference about WPV1 outbreaks and cVDPV2 outbreaks. As an aside, the figures on the WPV1 outbreak analysis are great – particularly Figures S15-S17.

The issue of immunity is a bit unclear if these are considered in the models directly, or used to explain deviations from the predicted model after the initial model is fit.

The authors exclude cases that do not extend the outbreak wavefront in their analysis of cVDPV2 cases. However, wouldn't this bias the outbreak velocity measures? For example, if there was an outbreak with many cases, but these were all geographically localized, i.e. within the same location, then this would have an outbreak velocity of 0. This seems to be important to consider, since this would provide a clearer picture of all the cVDP2 outbreaks. Further, excluding additional local cases removes information about the true spatial spread of the outbreak. For example, if there were several cases all local and then on day n there was an outbreak much further away, then this would be relevant information since it would show that there are differences in the outbreak velocity over the duration of the outbreak.

The authors primarily rely on physical distance between locations, however have the authors considered other measures such as travel time distance – of which there are publicly available measures – to help consider the local geography and road network?

The authors only consider reported cases of cVDP2, which is known for being underreported. Some analysis to consider the issues of sampling, particularly geographic biases, should be conducted.

[Added after correspondence with editor]: The authors calculate a Q statistic, that measures the difference in R2 values from a regression using environmental variables or one that only takes into account distance. This metric has been used in multiple phylogeographic analyses to explore the extent to which environmental distances (that take into account various spatial environmental rasters) over a null model. While this metric has been used in multiple peer reviewed manuscripts, (roughly 100 papers have cited the original Dellicour et al paper), it is not clear why this statistic is used instead of conducting a proper statistical model comparison. For example, it is well known that R2 increases with the addition of any explanatory variable (junk or otherwise), regardless of their importance or relationship with the outcome of interest. Further, for model comparisons, it is more common to use something like a likelihood ratio test, as a formal test, as opposed to comparing the summary measures or at minimum a summary statistic that accounts for the possibility of introducing unnecessary and unrelated predictors. If a full likelihood could not be constructed, then quasilielihood approaches could also be considered. Finally, when conducting model selection, maximizing the difference in R2 is not a robust way to choose which variables are the most scientifically relevant or predictive of to explain the variability in the outcome of interest.

Minor comments:

While the outbreak abbreviation names may be familiar to the authors, or those who work primarily in polio, to a general reader the naming conventions are not clear and difficult to follow. Is it possible to also provide the full names of these outbreaks (and use the abbreviated names for figures, tables, etc.)?

Figure 1 part C – could more distinct colors be used for the different regions to make it easier to tell which outbreaks had broader geographic expansion than others?

The authors state that, outbreaks have emerged in 19 different countries and spread to at least 22 countries, however you cannot tell this from the figure since the many points are clustered together and overlapping. Further, you cannot tell which outbreaks are associated with different countries from this plot. Additional figures that are clearer would be helpful to the reader.

How was the threshold of only considering outbreaks with 20 or more cases determined? Particularly since polio is highly underreported, do the authors have a sense of what estimated outbreak sizes they are excluding with this threshold? Or if the results hold with a lower threshold?

Figures S9 and S10, S12 are all incredibly difficult to read and should be expanded to make it easier to understand the results.

Reviewer #3 (Remarks on code availability):

For double blind reviews, I recommend not hosting the code on GitHub since that is often associated with a (very identifiable GitHub user). I could not run the code, because a number of files were missing, for example - Human_popD.tif, Outbreak_results/NA.csv. Further, no code is provided (or a workflow) to conduct the phylogenetic analysis on the GitHub link.

Version 1:

Reviewer comments:

Reviewer #1

(Remarks to the Author)

All of my comments have been addressed.

(Remarks on code availability)

Reviewer #2

(Remarks to the Author)

I believe that the authors have satisfactorily addressed the reviewers' comments and I am happy to recommend publication. I do not have further comments.

(Remarks on code availability)

N/A

Reviewer #3

(Remarks to the Author)

In this study the reviewers reconstruct spatial diffusion of wild type and vaccine derived polio outbreaks. The study offers an interesting natural experiment where the wild type virus has become rare and sporadic outbreaks are associated with control and eradication efforts in context of spread to neighboring countries. From the global dataset there is a limited set that allows the authors to propose population immunity could impact the rate of spread between countries. This paper presents a comprehensive analysis of existing data. Overall this paper is well written and provides a compelling argument for the value in sharing sequence data.

I have no major concerns. The initial review and response is fairly comprehensive. I found the study limitation section to be very valuable. Given the limitations and other possible explanations for the variable wavefront speed between countries, I suggest removing the reference to population immunity in the abstract, or at least soften the language.

Finally, Figure 3A is introduced at the end of the "Spatiotemporal dynamics of cVDPV2 spread" section. The rest of Figure three is in the "Wavefront velocity of cVDPV2 outbreaks and the impact of international borders" section. Fig 3D is discussed after Figure 3E. I think these figures could be discussed in the Wavefront section instead of splitting them between the two, and Figure 3E and 3D could be rearranged. I know this is nitpicking. You can take or leave this suggestion.

JBahl

(Remarks on code availability)

Decision Letter:

Our ref: NMICROBIOL-24123895A

29th August 2025

Dear Darlan,

Thank you for submitting your revised manuscript "Unravelling spatiotemporal heterogeneities of wild and vaccine-derived poliovirus spread: past and present" (NMICROBIOL-24123895A) and for your continued patience. It has now been seen by the original referees #1 and #2, as well as a new referee who replaced the original referee #3. Their comments are below. The reviewers find that the paper has improved in revision, and therefore we'll be happy in principle to publish it in Nature Microbiology, pending minor revisions to satisfy the referees' final requests and to comply with our editorial and formatting guidelines.

Thank you again for your interest in Nature Microbiology Please do not hesitate to contact me if you have any questions.

Sincerely,

Reviewer #1 (Remarks to the Author):

All of my comments have been addressed.

Reviewer #2 (Remarks to the Author):

I believe that the authors have satisfactorily addressed the reviewers' comments and I am happy to recommend publication. I do not have further comments.

Reviewer #2 (Remarks on code availability):

N/A

Reviewer #3 (Remarks to the Author):

In this study the reviewers reconstruct spatial diffusion of wild type and vaccine derived polio outbreaks. The study offers an interesting natural experiment where the wild type virus has become rare and sporadic outbreaks are associated with control and eradication efforts in context of spread to neighboring countries. From the global dataset there is a limited set that allows the

authors to propose population immunity could impact the rate of spread between countries. This paper presents a comprehensive analysis of existing data. Overall this paper is well written and provides a compelling argument for the value in sharing sequence data.

I have no major concerns. The initial review and response is fairly comprehensive. I found the study limitation section to be very valuable. Given the limitations and other possible explanations for the variable wavefront speed between countries, I suggest removing the reference to population immunity in the abstract, or at least soften the language.

Finally, Figure 3A is introduced at the end of the "Spatiotemporal dynamics of cVDPV2 spread" section. The rest of Figure three is in the "Wavefront velocity of cVDPV2 outbreaks and the impact of international borders" section. Fig 3D is discussed after Figure 3E. I think these figures could be discussed in the Wavefront section instead of splitting them between the two, and Figure 3E and 3D could be rearranged. I know this is nitpicking. You can take or leave this suggestion.

JBahl

Version 2:

Decision Letter:

6th October 2025

Dear Darlan,

I am pleased to accept your Article "Historical and current spatiotemporal patterns of wild and vaccine-derived poliovirus spread" for publication in Nature Microbiology. Thank you for having chosen to submit your work to us and many congratulations.

Authors may need to take specific actions to achieve compliance with funder and institutional open access mandates. If your research is supported by a funder that requires immediate open access (e.g. according to [a href="https://www.springernature.com/gp/open-science/plan-s-compliance"> Plan S principles](https://www.springernature.com/gp/open-science/plan-s-compliance) or the [a href="https://www.springernature.com/gp/open-science/us-federal-agency-compliance"> NIH public access policy](https://www.springernature.com/gp/open-science/us-federal-agency-compliance)) then you should select the gold OA route, and we will direct you to the compliant route where possible. Because authors warrant under our subscription licensing terms that they haven't committed to licensing any version of their article under a licence inconsistent with the terms of our agreement – including the applicable embargo period – publication under the subscription model isn't suitable for authors whose funders require no embargo.

An online order form for reprints of your paper is available at [a href="https://www.nature.com/reprints/author-reprints.html">https://www.nature.com/reprints/author-reprints.html](https://www.nature.com/reprints/author-reprints.html). All co-authors, authors' institutions and authors' funding

agencies can order reprints using the form appropriate to their geographical region.

Congrats again to you and your co-authors! I am looking forward to seeing your paper published.

With kind regards,

P.S. Click on the following link if you would like to recommend Nature Microbiology to your librarian
<http://www.nature.com/subscriptions/recommend.html#forms>

** Visit the Springer Nature Editorial and Publishing website at http://editorial-jobs.springernature.com?utm_source=ejP_NMicro_email&utm_medium=ejP_NMicro_email&utm_campaign=ejp_NMicro for more information about our career opportunities. If you have any questions please click [here](mailto:editorial.publishing.jobs@springernature.com).

Open Access This Peer Review File is licensed under a Creative Commons Attribution 4.0 International License, which permits use, sharing, adaptation, distribution and reproduction in any medium or format, as long as you give appropriate credit to the original author(s) and the source, provide a link to the Creative Commons license, and indicate if changes were made. In cases where reviewers are anonymous, credit should be given to 'Anonymous Referee' and the source.

Reviewers's comments in white

Answers in red

Edited text in blue

Reviewer #1 (Remarks to the Author):

The authors define the spatiotemporal patterns of spread for VDPV2 and WT1 poliovirus. The former is defined using available case data collected during surveillance for AFP cases, which includes locations and time. The latter is based on whole genome sequences. The major contribution is quantifying the rate of spread, primarily in several large outbreaks in Africa. These data can theoretically be used to better time and target vaccination campaigns to control outbreaks. More speculative, and less substantiated, is an analysis relating cross border spread to the levels of population immunity. This is interesting, but - as the authors indicate - an area that needs more investigation. With respect to their data on spread across borders, of the 76 outbreaks, only 8 spread to other countries.

Answer: We thank Reviewer 1 for the overview and for their comments. Regarding the “more speculative” analysis relating cross border spread to the levels of population immunity: indeed, given current limitations of the methods used for the landscape analysis of the wavefront velocity, we cannot test the direct effect of time-varying variables, such as population immunity. This is why, population immunity levels here are only presented as a potential explanation for the non-speculative and well-substantiated varying effect of international borders on the wavefront velocity of the two largest cVDPV2 outbreaks, i.e., a large negative effect of 38% for NIE-ZAS-1 vs a small positive effect of 3% for NIE-JIS-1. Given that both outbreaks started in the same region, northern Nigeria, at different times and spread to the same countries, the main immediate variable that could potentially explain the striking difference in the effect of borders is population immunity. What we show is that indeed, population immunity levels between the two outbreaks were very different across international borders and could potentially explain these differences, even though that cannot be formally tested as of. Because of that, we have been very open about the nature of our analysis of population immunity data as it can be seen on the exert from the main text below.

In light of this comment, we have now made the text clearer to explicitly state that we have not formally tested the effect of population immunity of the wavefront velocity, but it is a possible explanation of the varying effect of international borders between the two outbreaks at different points in time.

Lines 338-342

“In summary, these observations suggest that the resistance effect of international borders on the wavefront dispersal velocity could potentially be attributed to population immunity, although not formally tested here (see limitations), and that, the impacts of population density and accessibility seem to be considerably more modest and context dependent. For all effects tested, please see Table S2.”

Lines 437-442

“Finally, due to limitations in the software used for the wavefront analysis (see methods), which does not support time-varying covariates, population immunity could not be formally tested as a covariate using the wavefront velocity approach in our study. This is why spatiotemporal dynamics of immunity estimates has only been used as potential explanation for the effect of international borders and its potential role in poliovirus spread therefore warrants further investigation and formal testing.”

Overall, this builds on prior work looking at factors influencing emergence and spread of VDPV, a major issue in the polio eradication campaign. These prior studies are well cited and described. For example, one that link emergences of VDPV as associated with large campaigns in Nigeria, Ethiopia, and DRC in populations with low population immunity and no follow up campaigns.

This reviewer found that Q statistic a little hard to grasp. Given that it is important for the inference, suggest that this be specifically reviewed by others.

Answer: We understand that the Q statistic approach might cause some confusion as it corresponds to a relatively new analytical procedure. We have further detailed the related section in the Supplementary Methods, and please also see below our answer to a related comment by Reviewer #3. We would also like to refer the Reviewer to reference number 19 (see below), a study that was led by the second author of this manuscript, Simon Dellicour, and where this analytical framework was first described in much more detail (see also a visual summary of this analytical workflow taken from the study of Dellicour *et al.* and corresponding to Figure 2 therein).

19. Dellicour, S. et al. Unravelling the dispersal dynamics and ecological drivers of the African swine fever outbreak in Belgium. *J. Appl. Ecol.* 57, 1619–1629 (2020)

Figure 2 from Dellicour *et al.* (2020, *J. Appl. Ecol.*): analytical workflow implemented and applied to analyse the dispersal dynamics and drivers of the African Swine Fever virus (ASFV) outbreak in the wild boar population of southern Belgium (2018-2019). “SL”, “CS” and “LR” refer to “straight-line”, “Circuitscape” and “linear regression”, respectively. Circuitscape cumulative current maps here are log-transformed and obtained when connecting each infection case to the first detected case. These maps are thus illustrative of the Circuitscape path model used to compute the ecological distances involved in the linear regression analyses (similar tests were performed by considering the least-cost path model, see the text for further details). (*) Stochastic rotations of dispersal vectors were performed by respecting the proportion of filtered cases falling within (75.4%) or outside (24.6%) forest areas.

Major critiques

I am struggling to interpret the differences between Figs 3B and S11. The latter includes outliers and many of the outbreaks have extremely long tails of rates that appear to have large number of wavefront estimates (dots). With this level of skew - with many wavefront estimates of 100-400 km/day, what is one to think of the conclusions regarding differences in the 1-10 km/day range? Yes there are fast movements involved, but what does this mean for inferences regarding the low rates and the importance of the factors involved? Similarly, the authors claim that the wavefront can be used to target and time vaccination campaigns. Would one to that based on the median or the tail of the skew?

Answer: This is a good point that if the speed of spread is to be used to target campaigns, one should be cautious and capture an extent of the tail of the distribution. However, extremely fast wavefronts (100-400 km/day), are estimated to be proportionally much less frequent (0.02%)

than short distance movements . The 90%, 95%, 97.5% and 99% quantiles are 6.6, 9.2, 12.5, and 19.4 km/day, respectively (compared to the median wavefront velocity is of 2.3 km/day). Meaning that, over 95% of the movements are under 12.5 km/day and over 99% are under 20 km/day.

Of course, a formal analysis to inform vaccination campaigns would have to include the distribution of the wavefront velocity.

We would also like to highlight that we do not propose for the wavefront velocity to be used as the only tool to predict cVDPV2 movement. Other predictors, such as historical patterns of viral spread, which can be reconstructed with phylogenetic data and population immunity levels, should also be considered and are both described in this manuscript. However, the wavefront velocity gives a starting point to understand potential virus spread and the size of vaccination responses that need to be put in place, even in the absence of genetic data or population immunity levels.

That said, we have now modified the text to make it clearer about the suggested use of the wavefront velocity by surveillance and response programs and the use of quantiles to consider different case scenarios.

Lines 289-292

“Our results show that cVDPV2 outbreak wavefronts move at a median speed of 2.3 km/day and a 90% quantile of 9.2 km/day across most outbreaks, with specific outbreak median wavefront velocities ranging between 0.9-3.8 km/day.”

Lines 466-470

“Thus, future vaccination campaigns aiming to interrupt poliovirus transmission should proactively account for spread between neighboring regions, even in the absence of reported cases, and may use wavefront velocity estimates, including measures of variation (e.g., 90th percentile), to guide the geographical scope of interventions.”

The main “sell” of the manuscript - the relationship between population immunity and the impact of borders on rate of spread is based on comparison of two outbreaks with opposing impacts of borders. This seems tenuous.

Answer: The main sell of the manuscript is the use of different sources of data to understand the spatiotemporal patterns of poliovirus spread globally. We use WPV1 sequences to reconstruct historical movement patterns between countries globally and, in the absence of cVDPV2 sequences, we use geolocated case counts to estimate velocity of spread, which can also be used

to inform surveillance and response campaigns. Indeed, while we do not formally test the relationship between population immunity levels and a lower wavefront velocity, this is however introduced as a potential explanation for the differences in the impact of international borders on the cVDPV2 wavefront velocity, rather than formally establishing a relationship between population immunity and borders (please see the exert below where we make this explicit). Further methodological development would be required to assess the impact of time-varying covariates (e.g., population immunity) on the wavefront velocity. In short, to enable the use of time-series environmental layers, the Circuitscape algorithm would need to be adapted to work in three dimensions, allowing for the processing of successive environmental rasters over time.

We have changed the text to make it clearer how and why we use population immunity as a potential explanation for the opposing effect of international borders and add that to the limitation section of the manuscript.

Lines 437-442

“Finally, due to limitations in the software used for the wavefront analysis (see methods), which does not support time-varying covariates, population immunity could not be formally tested as a covariate using the wavefront velocity approach in our study. This is why spatiotemporal dynamics of immunity estimates has only been used as potential explanation for the effect of international borders and its potential role in poliovirus spread therefore warrants further investigation and formal testing.”

I agree with the authors that it is a major limitation that there weren't sequences available for VDPV2. These should exist, as the authors indicate. Why are they not available for inclusion in the analysis. This would seem to be important to verify that the outbreaks are ascertained correctly with respect to emergences and sources.

Answer: We strongly agree with the Reviewer that such sequences should be made available. This was one of the main limitations for our analyses but also one of the main reasons why this manuscript will have a large impact within the polio community and beyond, as it clearly outlines the need for analysis of poliovirus sequences from current outbreaks. Our results have been presented to the WHO, CDC and Gates Foundation and this has strengthened the case for sharing polio sequence data at least within the Global Polio Eradication Initiative (GPEI). A conversation has begun to share these data in future.

By way of background for the Reviewer: Isolates from every single case of polio have been sequenced since the early 2000s. In fact, vaccine-derived poliovirus can only be confirmed through genetic sequences to distinguish them from vaccine virus, and this is how the polio program associates cases to different emergences/outbreaks. The Poliovirus Information System (POLIS) at WHO, contains all epidemiological information on each one of the cases and some

information drawn from the genetic sequences, such as, number of mutations compared to Sabin poliovirus and the emergence group to which they belong, but not the sequences themselves. Sharing of genetic polio data even within the Global Polio Eradication Initiative has been very limited, which has, unfortunately, prevented further analysis from being carried out.

Poliovirus genetic data has only been made available through some publications, and publications on cVDPV2 outbreaks have been very scarce. Although a few cVDPV2 sequences have been made available (as one can see on Figure S13), most of them belong to outbreaks prior to the switch (2016), have not spread internationally, and/or have no metadata available. As such, we cannot reproduce the same global level analysis for cVDPV2 at this time, and this is why WPV1 historical sequences have been used to highlight the inference that can be made from such data. Also, in the absence of cVDPV2 sequences, we have performed the second-best analysis, which is estimating the wavefront velocity for each outbreak in the absence of information on the link between cases and investigating the impact of external factors on the dynamic of the wavefronts' progression. This scenario of data availability is the case for many other infectious disease outbreaks in several countries for which genetic information is not at all available and the wavefront analyses represents an alternative approach. Certainly, access to genetic sequence data is the best scenario but focusing on alternative methods when this is not the case is also crucial. We thank the Reviewer for their comment, and we have made this clearer in the limitations section.

Lines 419-425

“Given the limited number of cVDPV2 sequences publicly available since The Switch, our phylogeographic analysis focused on historical WPV1 sequences to shed light on similarities for current cVDPV2 spread. The Polio Information System (POLIS) contains information on outbreaks/emergence groups and mutation counts for each case but provides no access to the sequences themselves. Since all VDPV cases were sequenced since the early 2000s, analysis of such data would provide a more accurate insight into the current patterns of cVDPV2 spread.”

Minor critiques

Figure S8 is mislabeled S10 in supplement.

Answer: We thank the Reviewer for pointing this out. Labelling has now been corrected.

Figure S10 (the real one) - isn't entirely convincing as displayed in terms of relating speed of spread to the levels of population immunity. For example, NIE-SOS-7 and ANG-HUI-1 have different color patterns at the levers (ends of regression). And also looking at NIE-JIS-1, supports lines 149-152 in text.

Answer: Figure S10 shows the correlations between time and distance for each outbreak considering all reported cases. Each case is colored according to the estimated population immunity for their district of reporting at the time of reporting. What we show here is that for outbreaks with higher correlation between time and distance (>0.5), cases tended to be reported in areas of lower population immunity, with the opposite for outbreaks with lower correlation. The differences noted by the Reviewer are explained by the different levels of population immunity in those areas at the observed time. Population immunity is only one of multiple variables that can explain the specific differences between outbreaks, as we mention in the text that “population immunity can partially explain that...”. However, we are interested in the major observed effect of population immunity rather than the specifics at each time period, and this is why we focus on the Pearson’s r value as summary statistics for each outbreak. That said, to make our point clearer we have added a second supplementary figure which summarizes this relationship more appropriately (see below). Figure S10B shows the difference in the estimated population immunity for outbreaks with strong correlation (Pearson’s $r \geq 0.6$) when compared to other outbreaks (Pearson’s $r < 0.6$). Median population immunity for outbreaks with low or no correlation was almost double that for outbreaks with stronger correlation between distance and duration, 62.8 vs 33.8 respectively, and the difference is highly statistically significant p -value < 0.001 . As you would expect, this finding suggests population immunity disrupts the relationship between time and distance of spread as it becomes harder for the virus to encounter susceptible individuals.

Figure S10B - Comparison between the estimated admin-2-level population immunity for cVDPV2 outbreaks with strong correlation between duration and distance (Pearson's $r \geq 0.6$) vs other outbreaks (Pearson's $r < 0.6$). Poliomyelitis cVDPV2 cases and their respective estimated admin-2 population immunity at the time of symptom onset were discretized according to the level of correlation between distance from outbreak origin and observed duration of circulation (time of detection) for the outbreak (emergence group) they are linked to (see Figure S10A). Each dot represents a case and their associated population immunity. The boxplot depicts the interquartile interval (box) and median (large dot), whiskers show minimum and maximum value, and the violin plot shows the density of the data distribution.

Lines 248-250 Is it really striking that most spread happens between neighboring countries?

Answer: The lines the Reviewer refers to can be found below:

“Such patterns, although being from different types of polioviruses and circulating over a different time period are strikingly similar to patterns seen in the cVDPV2 AFP case data with most movement happening between neighbouring countries.”

We thank the Reviewer for their question. The word striking here refers to the fact that cVDPV2 case count spatiotemporal patterns are similar to those of historical WPV1 genetic sequences, and not to the fact that movement happens between neighboring countries. In addition, as WPV1 genetic data is historical (1958-2015), we were surprised that even though human connectivity has changed over time, patterns of spread are still very similar. In any case, the word striking has now been removed, without any real damage to the main argument.

Reviewer #3 (Remarks to the Author):

In “Unravelling spatiotemporal heterogeneities of wild and vaccine-derived poliovirus spread: past and present” the authors use cases of cVDPV2 (vaccine-derived cases of wild-type poliovirus 2) and sequences of WPV1 (wild-type poliovirus 1) to explore spatiotemporal patterns and drivers of polioviruses. The paper’s primarily focus is on cVDPV2 outbreaks, given the importance for recommended vaccination programs, current global spread, and geographically widespread outbreak risk. Overall, the authors conduct a thorough analysis of both cVDPV2 and WPV1 data, however the relationship between cVDPV2 and WPV1 outbreaks should be strengthened.

Major comments:

For each outbreak, is it possible to provide a table with the name (abbreviation), time the outbreak began (possibly just a month-year or year), what other countries this outbreak spread to, aggregated case counts for the countries, and duration of outbreak. I recognize that the authors state there are limitations to sharing the line list data, however, without additional information it is difficult to recreate the results presented or interpret them in context.

Answer: Thank you for the relevant suggestion. We now provide Table S1 with such information.

The WPV1 analysis is sound on its own, however the authors do not provide support that cVDPV2 would spread like WPV1 and hence that the WPV1 phylogenetic results would be relevant. Since there are cases and sequences in several of the countries they consider, a more direct comparison, i.e. showing that results from the analysis of cases of WPV1 are like the results identified for cases of cVDPV2 would provide a clearer link. For example, are the outbreaks of cVDPV2 and WPV1 of similar size? Geographic spread? Duration? Or, although there may not be sufficient samples, an analysis of the cVDPV2 sequences should be done. Even if the phylogenetic analysis of cVDPV2 sequences is quite limited, it would be helpful to bridge the gap between the inference about WPV1 outbreaks and cVDPV2 outbreaks. As an aside, the figures on the WPV1 outbreak analysis are great – particularly Figures S15-S17.

Answer: We thank the Reviewer for the comment and for the compliment to Figures S15-S17. Transmission of cVDPV2 has been established to be similar to that of WPV1

(<https://pubmed.ncbi.nlm.nih.gov/20573924/>). Unfortunately, we cannot perform the phylogeographic analysis of cVDPV2 outbreaks as sequences from cVDPV2 outbreaks that spread internationally are not available, as aforementioned when replying to the related comment of Reviewer #1; an aspect that has now been explicitly clarified in the text. However, we do present a comparison between the patterns of spread recovered from cVDPV2 case data and WPV1 genetic sequences, and which show that historic international WPV1 genetic spread resembles that of current international cVDPV2 spread (see also Figures 2 and 4).

Lines 419-425

“Given the limited number of cVDPV2 sequences publicly available since The Switch, our phylogeographic analysis focused on historical WPV1 sequences to shed light on similarities for current cVDPV2 spread. The Polio Information System (POLIS) contains information on outbreaks/emergence groups and mutation counts for each case but provides no access to the sequences themselves. Since all VDPV cases were sequenced since the early 2000s, analysis of such data would provide a more accurate insight into the current patterns of cVDPV2 spread.”

The issue of population immunity is a bit unclear if these are considered in the models directly, or used to explain deviations from the predicted model after the initial model is fit.

Answer: Population immunity is not directly considered in the wavefront analyses, as the approach used to investigate the impact of external factors on the dynamic of the wavefront progression cannot test time-varying variables, which is directly related to the nature of the algorithm (Circuitscape) used to compute the environmental distances associated with dispersal events (see also our answer to the related comment of Reviewer #1). Population immunity here is only used to contextualize the two outbreaks and as a potential explanation for the different observed impacts of borders in the wavefront velocity. We have now made the text clearer and to explicitly treat this aspect as a limitation.

Lines 338-342

“In summary, these observations suggest that the resistance effect of international borders on the wavefront dispersal velocity could potentially be attributed to population immunity, although not formally tested here (see limitations), and that, the impacts of population density and accessibility seem to be considerably more modest and context dependent. For all effects tested, please see Table S2.”

Lines 437-442

“Finally, due to limitations in the software used for the wavefront analysis (see methods), which does not support time-varying covariates, population immunity could not be formally tested as a covariate using the wavefront velocity approach in our study. This is why spatiotemporal dynamics of immunity estimates has only been used as potential explanation for the effect of international borders and its potential role in poliovirus spread therefore warrants further investigation and formal testing.”

The authors exclude cases that do not extend the outbreak wavefront in their analysis of cVDPV2 cases. However, wouldn't this bias the outbreak velocity measures? For example, if there was an outbreak with many cases, but these were all geographically localized, i.e. within the same location, then this would have an outbreak velocity of 0. This seems to be important to consider, since this would provide a clearer picture of all the cVDP2 outbreaks. Further, excluding additional local cases removes information about the true spatial spread of the outbreak. For example, if there were several cases all local and then on day n there was an outbreak much further away, then this would be relevant information since it would show that there are differences in the outbreak velocity over the duration of the outbreak.

Answer: We understand the Reviewer's point and would like to take the opportunity to clarify the difference between outbreak velocity and wavefront velocity. Outbreak velocity would correspond to a velocity estimate obtained when considering all cases occurring in the outbreak, i.e. even those that do not extend the area of the outbreak. On the other hand, the wavefront velocity estimated here focuses only on the spread of the outbreak into new areas, which is one of the novelties of the approach and the most relevant when planning the geographic scope of outbreak response campaigns. As such, cases which happened within the area already affected do not contribute to extending the outbreak geographically and would wrongfully bring the median/mean wavefront velocity of spread down.

Our manuscript, however, does also include a simpler regression analysis between time and distance of all cases, which gives us an idea of what outbreak velocity estimates could be (Figure S10), and for which the slope is 1.2 km/day. In any case, although lower, as it considers all cases, it is still close enough to the wavefront velocity estimated here.

We would like to refer the Reviewer to other studies that also only used cases that extended the wavefront to estimate a wavefront velocity (Kraemer 2019, Nat Microbiol; doi: 10.1038/s41564-019-0376-y, Zinsker 2015, Lancet Infect Dis; doi: 10.1016/S1473-3099(15)00234-0 and Tisseuil 2016, Ecography; doi: doi.org/10.1111/ecog.01393), one of which has found a very similar velocity of spread for the West Africa Ebola outbreak of 2.7 km/day (1,2). We have now further clarified the justification of the consideration of the wavefront velocity in the supplementary text.

The authors primarily rely on physical distance between locations, however have the authors considered other measures such as travel time distance – of which there are publicly available measures – to help consider the local geography and road network?

Answer: We thank the Reviewer for this relevant suggestion. We now also analyse an additional predictor consisting of pairwise measures of human transport connectivity obtained by computing pairwise distances on the accessibility (time to travel one meter, friction raster) generated and released by Weiss and colleagues (2018, *Nature*; doi: 10.1038/nature25181). Distances computed with Circuitscape on this raster correspond to a proxy of pairwise human connectivity among locations. Considering such a variable in our analytical framework thus allows us to investigate if human connectivity can constitute a better predictor of dispersal duration than geographic distance. In other words, we now also test if human connectivity measures can explain a significant proportion of the estimated variability in wavefront velocity. This additional analysis is now detailed in the revised version of our manuscript and the related results are reported in the text as well as in the updated version of Table S1. Overall, these additional analyses reveal that travel time to the nearest city can be a resistance factor as it is significantly associated with a reduction of 6% in the wavefront velocity of one of two main cVDP2 outbreaks (NIE-ZAS-1; see the updated version of Table S1). However, the actual impact of travel time, as measured by the raster by Weiss and colleagues, seems to be limited in magnitude and somehow conditional as it is not significant for NIE-JIS-1.

The authors only consider reported cases of cVDP2, which is known for being underreported. Some analysis to consider the issues of sampling, particularly geographic biases, should be conducted.

Answer: We thank the Reviewer as this is an important point to be mentioned in the limitation section of our manuscript. Yes, there are asymptomatic infections and there are gaps in AFP surveillance. Unfortunately, there are currently not reliable measures of case surveillance quality that could be used to address this limitation at a fine spatial scale. GPEI does measure surveillance quality through the non-Polio AFP rate, but this measure can mask surveillance gaps the subnational level. As such, the actual wavefront velocity may indeed be faster than the observed wavefront velocity as earlier infections might have been missed. This is also a potential explanation for the variation in the wavefront velocity observed among and between different outbreaks. However, the wavefront velocity estimated here still reflects what is currently observed by surveillance systems in all their complexities and as such provides a relevant and actionable real-world measure. In addition, we would expect the higher quantiles to contain the actual wavefront velocity, and our recommendation is that the quantiles are used to consider different case-scenarios when using wavefront velocity for decision making. Finally, if the cVDPV2 genomic data were to be made available, the likely time of introduction into new areas could be estimated and surveillance gaps could be inferred. Again, this reinforces the relevance

of this manuscript to push poliovirus genetic data to be shared more widely. We have now updated the limitation section to include such consideration:

Lines 433-437

“In addition, as surveillance systems are often affected by underreporting, the true wavefront velocity might be faster than that estimated here. However, the true median wavefront velocity is likely within the bounds of the 90% quantile interval reported here, which should be considered when projecting poliovirus spread (see discussion).”

Minor comments:

While the outbreak abbreviation names may be familiar to the authors, or those who work primarily in polio, to a general reader the naming conventions are not clear and difficult to follow. Is it possible to also provide the full names of these outbreaks (and use the abbreviated names for figures, tables, etc.)?

Answer: Yes, thank you for the suggestion, this has now been provided as part of the additional Supplementary Table aforementioned. In brief the outbreaks are names with three letter codes corresponding to country of first detection, first administrative area of first detection, and the number corresponding to how many outbreaks there have been in that location. They are not ever named in full, but we appreciated this is confusing to a non-polio audience.

Figure 1 part C – could more distinct colors be used for the different regions to make it easier to tell which outbreaks had broader geographic expansion than others?

Answer: Yes, thank you for the suggestion, we have now changed that figure accordingly.

The authors state that, outbreaks have emerged in 19 different countries and spread to at least 22 countries, however you cannot tell this from the figure since the many points are clustered together and overlapping. Further, you cannot tell which outbreaks are associated with different countries from this plot. Additional figures that are clearer would be helpful to the reader.

Answer: Thank you for your feedback, we have now improved the readability of those figures.

How was the threshold of only considering outbreaks with 20 or more cases determined? Particularly since polio is highly underreported, do the authors have a sense of what estimated outbreak sizes they are excluding with this threshold? Or if the results hold with a lower threshold?

Answer: As mentioned in the main text, most cVDPV2 outbreaks are very small, with a median of 4.5 cases per outbreak. This becomes clearer in the figure below, where 52.6% of the outbreaks (40/76) had only five reported cases or less. Since the wavefront velocity analyses only focuses on cases that extend the wavefront of the outbreak, most cases are eliminated from

the model leaving very few cases to work with and making it sometimes not possible to estimate the wavefront reliably for outbreaks smaller than 20 cases, as there is often not enough movement into new areas. In addition, for the hypotheses testing part of our study, we have also decided to focus on larger outbreaks as these are the ones for which response likely failed and are more representative of uncontained spread.

Distribution of outbreaks based on outbreak size, i.e., total number of reported cases in the period of study.

Figures S9 and S10, S12 are all incredibly difficult to read and should be expanded to make it easier to understand the results.

Answer: Thank you for the comment; the layout of those figures has been modified to improve their readability (see for instance the updated version of Figure S12 below).

A.

B.

Reviewer #3 (Remarks on code availability):

For double blind reviews, I recommend not hosting the code on GitHub since that is often associated with a (very identifiable GitHub user). I could not run the code, because a number of files were missing, for example - Human_popD.tif, Outbreak_results/NA.csv. Further, no code is provided (or a workflow) to conduct the phylogenetic analysis on the GitHub link.

Answers: As files larger than 50Mb cannot be hosted within GitHub, this is why the files above were missing from the repository. We have now updated the Readme section on Github with information on how to download the missing files. Regarding the phylogenetic analysis, it was not performed using code, it was performed using different software which are mentioned in the supplementary materials with a detailed workflow of the analysis. We would be happy to include the same information in the repository if needed.

[Added after correspondence with editor]: The authors calculate a Q statistic, that measures the difference in R^2 values from a regression using environmental variables or one that only takes into account distance. This metric has been used in multiple phylogeographic analyses to explore the extent to which environmental distances (that take into account various spatial environmental rasters) over a null model. While this metric has been used in multiple peer reviewed manuscripts, (roughly 100 papers have cited the original Dellicour et al paper), it is not clear why this statistic is used instead of conducting a proper statistical model comparison. For example, it is well known that R^2 increases with the addition of any explanatory variable (junk or otherwise), regardless of their importance or relationship with the outcome of interest. Further, for model comparisons, it is more common to use something like a likelihood ratio test, as a formal test, as opposed to comparing the summary measures or at minimum a summary statistic that accounts for the possibility of introducing unnecessary and unrelated predictors. If a full likelihood could not be constructed, then quasiliikelihood approaches could also be considered. Finally, when conducting model selection, maximizing the difference in R^2 is not a robust way to choose which variables are the most scientifically relevant or predictive of to explain the variability in the outcome of interest.

Answers: The Reviewer is perfectly right when stating that the addition of any additional predictive variables – even “junk”/non-informative ones – will lead to a certain increase of R^2 . However, it is very important to note here that this does not correspond to what is performed in the analysis. Indeed, we do not conduct multivariate regression analyses nor perform regression model comparisons. Instead, we rather directly compare the amount of variability of a response variable, the dispersal duration (t), that can be explained by an environmental distance (d_{env}), i.e. a measure of spatial distance weighted according to the spatial heterogeneity of an underlying environmental layer, as opposed to a simple geographical distance (d_{geo}). As re-detailed below,

the aim of this procedure is to adopt a univariate regression approach to test if the heterogeneity of the dispersal velocity could be better explained when considering an environmental distance instead of a geographic distance. Therefore, we are not comparing different multivariate regression models but directly aim to assess and test whether d_{env} is more correlated to t than d_{geo} is.

The log-likelihood ratio test suggested by the Reviewer is only applicable when comparing nested models like, for instance, multivariate linear regression models where one corresponds to a special case of the other (e.g., $y \sim x_1 + x_2$ vs $y \sim x_1 + x_2 + x_3$). Here, on the other hand, we compare two distinct univariate regressions, which are based on the same response variable (dispersal duration t) but two different predictive variables (the environmental distances d_{env} and the geographic distance d_{geo}). Those two linear regression models thus differ by the considered predictive variable and are therefore not suited for a log-likelihood ratio test.

As now explicitly stated in the text, the choice of a univariate approach is here motivated by the very nature of the computed environmental distances. To compute such environmental distances, we used the path model available in the Circuitscape program, which implements an algorithm based on circuit theory. This algorithm approximates an environmental distance as the global electric resistance connecting two ‘nodes’ on an environmental raster treated as a grid of resistance (or conductance) values (McRae 2006, *Evolution*; doi: 10.1554/05-321.1). This algorithm presents the advantage of integrating the uncertainty in the path taken to connect two points on the map, as opposed to the simple geodesic distance between the same two points. It is important to keep in mind that the computed environmental distances are still correlated with the spatial distance between any pairs of spatial points. The resulting collinearity between environmental and geographic distances prevents the interest of meaningful multivariate analyses that would, e.g., attempt to predict the variability of the dispersal durations t according to geographic distance d_{geo} * and one or several environmental distances d_{env} computed on distinct environmental rasters ($t \sim d_{geo} + d_{env,1} + d_{env,2} + \dots$). Consequently, the approach adopted here (and which is indeed inspired by previous works in molecular epidemiology), rather consists in assessing if a higher proportion of the variability of t can be explained by a d_{env} as compared to d_{geo} using two separate univariate approaches.

In the original method development, several correlation metrics were explored but we eventually focused on the coefficients of determination also for its readability. Specifically, we estimate the statistic Q defined as the difference between two coefficients of determination: (i) R^2_{geo} estimated from the linear regression between the dispersal durations and the geographic distances ($t \sim d_{geo}$) and (ii) R^2_{env} estimated from the linear regression between the dispersal durations and environmental distance ($t \sim d_{env}$). This statistic Q is thus a straightforward and easily interpretable metric allowing a direct measure of any additional dispersal duration variability that could be explained when considering an underlying environmental heterogeneity.

Finally, please note that the level of significance of the coefficients of determination or of the Q statistic are not assessed using Fisher tests, which are usually conducted when we can assess that regression residuals are normally distributed and associated with a homoscedasticity. In our analytical procedure, the statistical support of Q is obtained by conducting a one-tailed test based on a null dispersal model generated through a randomisation procedure (i.e. a model mimicking the observed dispersal dynamics but in which the tested environmental layer did not have any impact on the dispersal velocity). While other correlation metrics or alternative analytical procedures could potentially be considered, the selection of this approach implemented in our study has been motivated by its tractability and the possibility to exactly formulate the hypotheses on the environmental impact on dispersal that we aim to test.

We recognise that those aspects remain relatively technical, and we have now improved and completed the related part of the text to enhance its clarity when explaining the rationale of the present analytical procedure.

(*) In our analytical framework, d_{geo} is not computed as the great-circle (or “geodesic”) distance between points on the map, but using the Circuitscape algorithm ran on a uniform “null” raster where the values of all accessible cell have been set to “1”. By doing so, the uncertainty in the path taken is also considered when computing a proxy for pairwise geographic distances; a proxy that is thus directly comparable to environmental distances computed with the same algorithm.

Reviewer #2 (Remarks to the Author):

The manuscript investigates the spatiotemporal spread of poliovirus, focusing on differences between wild poliovirus type 1 (WPV1) and circulating vaccine-derived poliovirus type 2 (cVDPV2). Using wavefront velocity models, phylogeographic analysis, and epidemiological data, this study examines factors influencing outbreak size and speed of transmission. The findings highlight the role of vaccination coverage, population immunity, and geographic barriers in shaping virus dispersal patterns.

This study is overall of high quality and provides important insights for poliovirus surveillance and outbreak response strategies, with implications for global eradication efforts. The methodology is robust, comprehensive and in parts innovative. It is clearly described and provides sufficient level of detail, with maybe a couple of exceptions (see minor comments below). The results are also clearly reported and the choice of figures adequate. The discussion is strong. If anything, it could maybe better highlight policy implications in this section. It would be useful, for instance, to indicate in more detail how these results are translatable and who policymakers could act on the findings of this study.

Minor comments

1. Page 5, Figure 2: The colour scheme used in this figure makes the dots difficult to distinguish, especially in panels A and B. It is also difficult to distinguish the different shapes in panel D, due to their small size. The authors should consider revising this.

It has now been revised.

2. Page 6, Figure 3, panel A: on the Y axis, 'outbrak' should be 'outbreak'.

It has now been corrected.

3. Page 6, Figure 3, panel D: should 'Côte d'Ivoire' be 'Ivory Coast'? Or, if the countries are listed in their official language, should 'Guinea' be 'Guinée'?

We thank the reviewer for their comment. We have been using location names, including countries, according to their WHO names.

4. Line 536: More detail on the method used to identify sequences with potentially excessive mutations or wrong collection dates on TempEst v.1.5.3 should be provided.

We thank the Reviewer for their comment. We used the residuals of the root-to-tip regression of genetic diversity against time of collection to investigate whether sequences could potentially have excessive mutations or a wrong collection date. Only one sequence raised suspicion as it was the only one to present a residual >0.2 , accession number HQ286311. The collection date on GenBank is 2009 but, however, the number of mutations is suggestive of a much earlier collection date. Indeed, the sequence name, NO9511238, which normally follows a pattern in which the first two numbers are the collection year, is suggestive of a collection date in 1995, compatible with the evolutionary rate of polio. Yet, as the correct sequence collection date could not be confirmed with the submitting group, this sequence was removed. In any case, this sequence would not have influenced any of the main analysis in this manuscript as it was collected in Indonesia and did not cluster within clades 2 or 4.

The removal of sequence HQ286311 was made clear in the methods text, as follows:

Lines 809-812

“One sequence (HQ286311) was excluded due to a high residual value, only sequence with residual $> \pm 0.2$, which is indicative of a potential error in the reported collection date. As the correct collection date could not be confirmed, the sequence was removed from the dataset.”

5. Page 9, Figure 4, panel A: the branch supports / posterior probabilities of the branches should be shown on the phylogenies, one way or another. I appreciate this may make the trees less readable, but it is necessary for the interpretation of the phylogenies. I would also increase the size of panel A relative to panel B (the map is useful but probably less informative than the tree itself).

We thank the Reviewer for their comment. We have added the posterior support to the main nodes which are relevant for the Clade 2 subtree. For the main tree, it would make it extremely difficult to read to add information on node supports. However, we have now added the Supplementary Figure S19 with a larger version of the tree in Figure 4 in which the support for every single node is made available. Finally, we believe that panel B is a rather important part of Figure 4 as the article is also aimed at public health/policy officials within and beyond the polio community which are normally not familiar with phylogenetic trees and for which the map conveys more information. We hope that the addition of Supplementary Figure S19 (separately submitted as too large to include here) will provide enough detail to readers who would be more interested in the phylogeny.

6. Line 554: The authors have adopted an uncorrelated relaxed molecular clock for all analyses. It would be useful to justify this choice of model, as it is likely to influence the inference of temporal patterns in this study. Was a strict molecular clock also tested against the datasets? If so, how?

We thank the Reviewer for their comment. As part of our initial exploratory analyses, we ran both an analysis based on a strict molecular clock and an analysis based on an uncorrelated lognormal relaxed clock model (UCLD). The strict clock is a special (nested) case of the UCLD, where the standard deviation and coefficient of variation of the evolutionary rate is equal to or includes zero, implying no rate variation among branches, as such the UCLD would allow for a strict clock and no rate variation was supported by the data.

Following this principle, we have, in fact, used a UCLD model for datasets A and B and a strict clock for datasets C-E, as you can see in the supplementary table below. For instance, in our UCLD analysis of dataset A, the estimated standard deviation of the mean rate (uclid.stdev) ranged from 0.006 to 0.01, and the coefficient of variation (coefficientOfVariation) ranged from

0.5 to 0.8, neither of which includes zero. These values reject the strict clock assumption and support the use of a relaxed clock model (https://beast.community/workshop_model_selection, <https://taming-the-beast.org/tutorials/Molecular-Dating-Tutorial/>). In particular, the coefficient of variation indicates substantial rate heterogeneity, with branch-specific rates varying between at least half to nearly equal the mean rate.

This has now been included in the methods section and the table below has been included as supplementary table 3.

“Time-rooted phylogenies were inferred using BEAST v.1.10.5⁵⁴ under a GTR+I+G substitution model and an uncorrelated relaxed molecular clock for datasets A and B and a strict clock for datasets C-E. Molecular clock selection was based on the standard deviation and coefficient of variation for the UCLD rates for each dataset and whether they included/abuted zero or not (https://beast.community/workshop_model_selection, <https://taming-the-beast.org/tutorials/Molecular-Dating-Tutorial/>). For details see Supplementary Table 6.”

Table S6. Information on clock models used for each dataset in the study.

Dataset	UCLD rate stats mean (95% HPD)		Final clock model
	Standard deviation	Coefficient of variation	
A - Global	7.6x10 ⁻³ (6.5-8.8x10 ⁻³)	0.66 (0.6-0.72)	UCLD
B - Clade 3	4.1x10 ⁻³ (2.7-5.6 x10 ⁻³)	0.391 (0.28-0.51)	UCLD
C - Clade 4a	1.2x10 ⁻⁴ (4.9x10 ⁻¹¹ -3.55x10 ⁻⁴)	2.0x10 ⁻² (4.09x10 ⁻¹⁵ -0.06)	Strict
D - Clade 4b	8.1x10 ⁻⁴ (1.9x10 ⁻⁴ -1.41x10 ⁻³)	0.1 (0.02-0.17)	Strict
E - WPV3	8.8x10 ⁻⁵ (2.1x10 ⁻⁸ -2.5x10 ⁻⁴)	1.2x10 ⁻² (3.3x10 ⁻⁶ -0.03)	Strict

Reviewer #2 (Remarks on code availability):

I do not have the expertise required to review the code provided with the manuscript.

Response to 2nd round of Reviewers' comments to manuscript NMICROBIOL-24123895

Reviewers's comments in white

Answers in red

Changed text in blue

Reviewer #3:

Remarks to the Author:

In this study the reviewers reconstruct spatial diffusion of wild type and vaccine derived polio outbreaks. The study offers an interesting natural experiment where the wild type virus has become rare and sporadic outbreaks are associated with control and eradication efforts in context of spread to neighboring countries. From the global dataset there is a limited set that allows the authors to propose population immunity could impact the rate of spread between countries. This paper presents a comprehensive analysis of existing data. Overall this paper is well written and provides a compelling argument for the value in sharing sequence data.

I have no major concerns. The initial review and response is fairly comprehensive. I found the study limitation section to be very valuable. Given the limitations and other possible explanations for the variable wavefront speed between countries, I suggest removing the reference to population immunity in the abstract, or at least soften the language.

We thank the reviewer for the comments and for the suggestion. We understand the reviewer has suggested to remove the mention to immunity in the abstract. However, the sentence becomes misleading when removing immunity as we have found that borders ONLY are significant when in the presence of high population immunity and saying that borders reduce spread would lead to a generalisation that is not corrected according to our own findings. We have tried to soften the language (highlighted in yellow).

Abstract – lines 27-39

“Outbreaks of vaccine-derived poliovirus type 2 (cVDPV2) have become a major threat to polio eradication. However, variations in spatiotemporal spread have not been quantified. We analyzed cVDPV2 cases and wild poliovirus 1 (WPV1) sequences to uncover spatiotemporal patterns and drivers of poliovirus spread. Between May 1, 2016 and September 29, 2023, 3120 cVDPV2 poliomyelitis cases were reported across 76 outbreaks in 39 countries. Outbreaks have median observed circulation of 202 days (0-1905) and median maximum distance of 231 km (0-4442). Wavefront velocity analysis of large outbreaks reveals median velocity of spread of 2.3 km/day (1.0-4.4). International borders are associated with slower velocity of spread ($p < 0.001$), in periods with high estimated population immunity. Phylogeographic analysis of 1572 global WPV1 sequences reveals that historic spread resembles recent cVDPV2 patterns and that international spread is largely sustained by unidirectional movement between neighbouring countries. Our findings offer insights for enhancing the geographical scope of vaccination response in the final phases of poliovirus eradication.”

Finally, Figure 3A is introduced at the end of the "Spatiotemporal dynamics of cVDPV2 spread" section. The rest of Figure three is in the "Wavefront velocity of cVDPV2 outbreaks and the

impact of international borders" section. Fig 3D is discussed after Figure 3E. I think these figures could be discussed in the Wavefront section instead of splitting them between the two, and Figure 3E and 3D could be rearranged. I know this is nitpicking. You can take or leave this suggestion.

We thank the reviewer for the helpful suggestion. Regarding Figures 3D and 3E (now 4D and 4E), the issue was due to a typographical error, which has now been corrected. Figure 3D is properly cited before Figure 3E (see below). With respect to Figure 3A, we acknowledge that it is discussed in a different section; however, we believe it would be out of place to introduce the wavefront velocity section with an analysis that is not directly related to this topic. Since Figure 3A addresses the broader relationship between time and distance, it is appropriately included within the same figure set, providing context and setting the stage for the subsequent panels on wavefront velocity and the relationship with borders and immunity. As such, we would prefer not to change the figure, as other reviewers also did not mention this as an issue.

Lines 266-277

“This is supported by a median wavefront velocity which is 32% faster for the NIE-JIS-1 outbreak (Fig. 4B) and for a faster invasion velocity for NIE-JIS-1 in 87.5% (7/8) of the countries that reported both outbreaks (Fig. 4D).

A possible explanation for these contrasting results is that most of the NIE-JIS-1 outbreak happened in 2020, when the median estimated immunity in children aged 6-36 months¹⁸ was high in Nigeria (median admin-2 immunity at symptom onset: 72.6%) but very low in all other countries involved in the outbreak (0.3%). Conversely, for the NIE-ZAS-1 outbreak, immunity was initially low in Nigeria and high in all other countries (in response to the ongoing NIE-JIS-1 outbreak). While immunity in Nigeria increased to high levels (84.8%), immunity in all other countries (65.5%) never decreased to levels as low as those during the NIE-JIS-1 outbreak (Fig. 4E).”

Fig. 4.

Spatiotemporal characteristics and drivers of cVDPV2 outbreaks. **(A)** Correlation between time (difference in days between date of paralysis onset each case and for case with highest distance from outbreak origin and that of the first case for each outbreak) and distance (distance from outbreak origin) for all cVDPV2 cases of large outbreaks (20 or more cases). Dotted line and shaded area represent the estimated linear regression line and the 95% confidence interval (two-sided test). Correlation was assessed using a two-sided Pearson correlation test. **(B)** Box

plot showing the distribution of wavefront velocity estimates for large cVDPV2 outbreak: NIE-ZAS-1 (n=578), NIE-JIS-1 (428), PAK-GB-1 (393), RDC-MAN-3 (340), YEM-TAI-1 (221), CHA-NDJ-1 (220), AFG-NGR-1 (135), RDC-KAS-3 (105), ANG-HUI-1 (79), ANG-LUA-1 (49), SOM-BAN-1 (40), NIE-SOS-7 (35), RDC-SAN-1 (32), ETH-ORO-1 (28), RDC-HLO-1 (27), RDC-MAN-5 (26), RDC-BUE-1 (24), RDC-HLO-2 (20). The central horizontal line indicates the median (50th percentile). The bounds of each box represent the interquartile range (IQR, 25th–75th percentiles). Whiskers extend to the most extreme data points within $1.5 \times \text{IQR}$ from the lower and upper quartiles. Whisker ends correspond to the minimum and maximum non-outlier values. Outliers were removed for ease of visualisation. Full version can be seen in Fig. S11. (C) Impact of international borders on the wavefront velocity of the two largest cVDPV2 outbreaks to date, NIE-ZAS-1 (578, ongoing, $p < 0.01$) and NIE-JIS-1 (428, $p = 0.01$). Red dotted lines represent the actual estimated Q for each outbreak. Q represents the proportion of the heterogeneity in the wavefront velocity that can be associated with the tested variable (see the Supplementary Materials for full details). Density plots depict the distribution of Q values obtained under a null dispersal model in which the tested environmental factors do not impact the outbreak wavefront velocity. As detailed in Supplementary Materials, such a null dispersal model being obtained by a stochastic rotation of dispersal vectors and tested using a one-sided permutation test. (D) Fold change of the invasion velocity of NIE-ZAS-1 over NIE-JIS-1. Invasion velocity for each outbreak was estimated using the geodesic distance and the time of onset for the first case in each country from each outbreak origin. (E) Median (line) and IQR (shaded areas) for the estimated vaccine-induced type 2 population immunity at the second administrative level and timeseries of reported cVDPV2 AFP cases for Nigeria and all other countries involved in the NIE-JIS-1 and NIE-ZAS-1 outbreaks, respectively. Lines and bars are coloured according to location, Nigeria (red) and other countries reporting cases from the same outbreak (blue). For details on immunity estimates, please refer to the supplementary materials. (F) Time lag (days) between the first case (date of onset) and the last vaccination campaign (campaign start date) per district for each outbreak according to country group: NIE-JIS-1 Nigeria (n=38), NIE-ZAS-1 Other (104), NIE-ZAS-1 Nigeria (192) and NIE-JIS-1 Other (221). The central horizontal line indicates the median (50th percentile). The bounds of each box represent the interquartile range (IQR, 25th–75th percentiles). Whiskers extend to the most extreme data points within $1.5 \times \text{IQR}$ from the lower and upper quartiles. Whisker ends correspond to the minimum and maximum non-outlier values.